# Intra-class Mixup for Out-of-Distribution Detection

## Abstract

Deep neural networks have found widespread adoption in solving image recognition and natural language processing tasks. However, they make confident mispredictions when presented with data that does not belong to the training distribution, i.e. out-of-distribution (OoD) samples. Inter-class mixup has been shown to improve model calibration aiding OoD detection. However, we show that both empirical risk minimization and inter-class mixup create large angular spread in latent representation. This reduces the separability of in-distribution data from OoD data. In this paper we propose *intra-class mixup* supplemented with *angular margin* to improve OoD detection. Angular margin is the angle between the decision boundary normal and sample representation. We show that intra-class mixup forces the network to learn representations with low angular spread in the latent space. This improves the separability of OoD from in-distribution examples. Our approach when applied to various existing OoD detection techniques shows an improvement of **4.68%** and **6.08%** in AUROC performance over empirical risk minimization and inter-class mixup, respectively. Further, our approach aided with angular margin improves AUROC performance by **7.36%** and **9.10%** over empirical risk minimization and inter-class mixup, respectively.

## 1 Introduction

Deep Learning has been employed by many state of the art machine learning models to effectively solve image recognition (Krizhevsky et al., 2012; LeCun et al., 2015) and natural language processing (Andor et al., 2016) tasks. Despite their effectiveness, recent research has shown the existence of inputs (Goodfellow et al., 2015b; Nguyen et al., 2015) that lead these networks to make confident mispredictions. Two classes of such inputs have emerged in literature. The first class, adversarial examples (Bruna et al., 2014; Goodfellow et al., 2015a; Kurakin et al., 2017), are specifically crafted inputs with the intent of fooling deep neural nets. The second class dubbed Out-of-Distribution (OoD) (Nguyen et al., 2015) examples are examples that do not belong to the underlying true distribution that the training dataset is drawn from. High confidence of the networks on such OoD examples makes it difficult to identify false classifications and poses a challenge to their deployment in safety critical scenarios (Amodei et al., 2016), from medical diagnosis to self driving applications.

To address this problem several methods have been proposed which aim to identify out-of-distribution samples. In the context of deep learning, OoD detection methods can be broadly classified into two categories, generative and discriminative methods. Generative methods use various techniques such as Gaussian Discriminant Analysis (Lee et al., 2018b), Generative Adversarial Networks (GANs) (Deecke et al., 2018; Lee et al., 2018a; Ren et al., 2019) or more generic generative methods (Wang et al., 2017) to model the underlying distribution and separate in-distribution samples from OoD samples. On the other hand, discriminative methods (Hendrycks & Gimpel, 2017; Hendrycks et al., 2019; Hsu et al., 2020; Liang et al., 2018; Liu et al., 2020) detect OoD samples based on measures such as softmax probability (Hendrycks & Gimpel, 2017; Hendrycks et al., 2019), energy scores (Liu et al., 2020) or other metrics computed from the trained network. Most of these metrics are a proxy for the distance of the sample representation from the decision boundary ($l_2$ margin in Figure 1a). Further, we show that existing techniques such as empirical risk minimization (ERM) and inter-class mixup (Zhang et al., 2018) result in networks learning representations that have large spread. This large spread creates broad overlapping representations for in-distribution and OoD data, making it hard to separate in-distribution data from OoD data. Additionally, the use of metrics that are proxies

for distance from decision boundary have a disadvantage when OoD points have similar margin as in-distribution data. For example, consider point "(b) OoD" in Figure 1a where the OoD data point is far from the class representation and has similar $l_2$ margin as in-distribution data; this OoD point will have similar OoD metric scores as in-distribution data. Using angular margin would help separate OoD data in such a case. To account for these factors, we propose intra-class mixup supplemented with angular margin (refer to Figure 1a) to improve OoD detection where angular margin is the angle between the normal to the decision boundary and the sample representation.

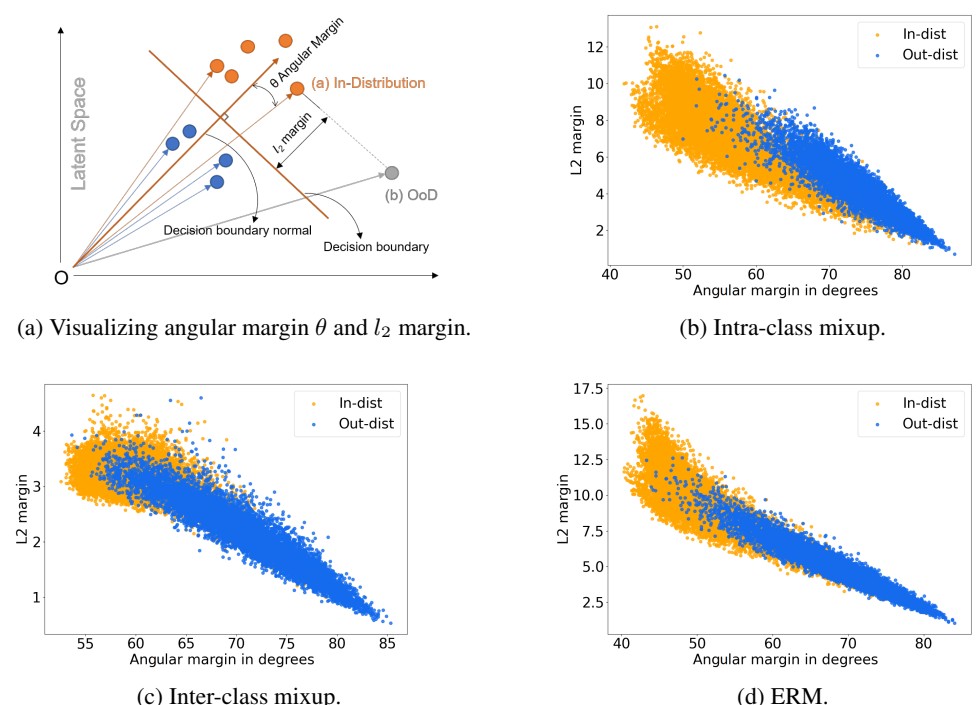

(a) Visualizing angular margin $\theta$ and $l_2$ margin.

(b) Intra-class mixup.

(c) Inter-class mixup.

(d) ERM.

Figure 1: Visualizing improved OoD separability of intra-class mixup trained ResNet18 model. SVHN was the in-distribution dataset and TinyImageNet was used as OoD dataset. For all the representations un-mixed inputs were used.

We show that the proposed intra-class mixup forces the network to learn latent representations with low variance (i.e. small spread), thus improving in-distribution and OoD separability. This is visualized in Figure 1, which plots the angular margin vs. $l_2$ margin for in-distribution and OoD data for ERM, inter-class mixup and intra-class mixup. Further, we show that intra-class mixup trained, angular margin augmented OoD detector achieves on average **7.36%** and **9.10%** improvement in AUROC performance over ERM and inter-class mixup, respectively.

In summary the contributions of this paper are as follows:

- We propose *intra-class mixup* to improve OoD detection. We show that intra-class mixup learnt latent space representations have lower angular spread when compared to ERM and inter-class mixup, thus, improving separability of OoD examples. We attribute this to the reduced input variance resulting from intra-class mixup.

- We show existing discriminiative OoD detection methods that use proxies for distance from the decision boundary benefit from intra-class mixup when compared to ERM and inter-class mixup.

- We leverage better angular separability to improve OoD detection. We show that supplementing angular margin with existing OoD detection scores improves OoD detection performance on various existing techniques.

## 2 RELATED WORK

In the context of deep-learning, the approaches to detect OoD examples can be broadly classified into two categories discriminative and generative methods. Under discriminative methods recent works (Hendrycks & Gimpel, 2017; Hendrycks et al., 2019; Hsu et al., 2020; Liang et al., 2018) have leveraged softmax probability scores to detect anomalous examples. The authors of Hendrycks & Gimpel (2017) presented their results on using softmax scores to detect OoD examples on several machine learning tasks. The idea of utilizing softmax scores was further extended in Hendrycks et al. (2019) by modifying the standard loss function used to train the classifier. The network was trained on an in-distribution dataset as well as an additional auxiliary or outlier dataset. The authors claim that such outlier exposure enables the detector to generalize and detect unseen anomalies. Further, the authors also observed characteristic properties of the auxiliary dataset which help improve detection performance. Another approach that leverages softmax scores are the ODIN (Hsu et al., 2020; Liang et al., 2018) techniques, which showed improvement to OoD detection without having to explicitly tune the out-of-distribution dataset which the authors argue is generally hard to define *a priori*. The authors proposed decomposing the confidence score and modifying the input pre-processing method. Other recent work under the discriminative techniques proposed the use of energy score (Liu et al., 2020) instead of softmax scores arguing that softmax based approaches suffer from overconfident posterior distributions for out-of-distribution samples. They claim that energy scores are theoretically aligned with the probability density of the inputs and is therefore less likely to result in overconfident predictions. Further, in a non deep learning context angular distance based approaches (Kriegel et al., 2008) have been successfully used in literature to avoid the curse of dimensionality.

Generative methods attempt to model the underlying true distribution and hence, identify samples outside the modelled distribution. The authors of Lee et al. (2018b) proposed a technique that modelled a class conditional Gaussian distribution with respect to features of the deep network and used a Mahalanobis distance based confidence score calculated from the Gaussian model to detect OoD examples. In Lee et al. (2018a) the authors proposed the use of Generative Adversarial Networks (GANs) (Goodfellow et al., 2014) to generate OoD samples to improve OoD detector training. They proposed two additional loss terms that were added to the original loss function. One of the proposed terms forced the classifier to be less confident on OoD samples while the other helped generate more effective OoD examples. Another work that investigated a generative model based approach for OoD detection was Ren et al. (2019) which proposed a likelihood ratio method which corrected for background statistics and achieved competitive performance on vision and sequence classification tasks. Approaches under both categories have focused on networks trained using Empirical Risk Minimization (ERM) which has been observed to result in undesirable behaviors such as memorization (Zhang et al., 2016) and sensitivity outside the training examples (Bruna et al., 2014). Inter-class mixup (Zhang et al., 2018) has been shown to be effective against such undesirable behaviours (Zhang et al., 2018; Thulasidasan et al., 2019) and has been shown to improve calibration and OoD performance (Thulasidasan et al., 2019). However, we show that inter-class mixup and ERM trained networks learn representations with large variance (i.e. large spread), impacting their OoD detection performance. To counter this effect, in this paper we propose intra-class mixup aided with angular margin for improving OoD detection.

## 3 BACKGROUND

**In-distribution.** The training and testing samples used for a machine learning task are drawn from some underlying distribution, also called the true distribution. The samples from this underlying true distribution are called in-distribution samples and are represented by $D_{in}$ in this work. However, in most practical learning scenarios the true distribution is unknown *a priori* and hence, the training distribution, $D_{train}$, is used as a proxy for $D_{in}$.

**Out-of-distribution.** A sample is said to be Out-of-distribution (OoD) if it does not belong to the underlying true distribution $D_{in}$, and is denoted by $D_{out}$. Since we do not have access to the true underlying distribution, for our experiments we use uniform noise, Gaussian noise and samples from other datasets for similar tasks as a proxy for Out-of-distribution samples or $D_{out}$. This is common practice in literature (Hendrycks & Gimpel, 2017; Hendrycks et al., 2019; Hsu et al., 2020; Lee et al., 2018b; Liang et al., 2018).

**Empirical Risk Minimization (ERM)**. The objective of learning algorithms, when the true underlying distribution is known, is to learn an optimal mapping $f(.)$ from the input $\boldsymbol{x}$ to an output $y$, where $(\boldsymbol{x}, y) \sim D_{in}$. This objective is satisfied by learning a function $f(.)$ that minimizes a loss function $\mathcal{L}(f(\boldsymbol{x}), y)$ over the entire distribution. We can compute the expectation of the loss function as shown in Equation 1. This is know as expected risk.

$$R_{\text{expected}} \quad = E_{\boldsymbol{x}, y \sim D_{in}}[L(f(\boldsymbol{x}), y)] \tag{1}$$

As in most realistic scenarios, we do not have access to the true data distribution, and hence, it is common to use available empirical data as proposed in Vapnik (1998) to approximate the expected risk. This form of approximation, presented in Equation 2, is know as Empirical Risk Minimization (ERM).

$$R_{\text{empirical}} \quad = \frac{1}{N_e} \sum_{i=1}^{N_e} \mathcal{L}(f(\boldsymbol{x}_i), y_i) \tag{2}$$

where $(\boldsymbol{x}_i, y_i) \sim D_{train}$ and $N_e$ is the number of training samples in $D_{train}$.

## 4 METHODOLOGY

### 4.1 ANGULAR MARGIN

In this subsection we define angular margin. Angular margin $\theta$ can be described by the following equations:

$$\boldsymbol{n} = \frac{W_c^{\hat{y}}}{|W_c^{\hat{y}}|} \tag{3}$$

$$\theta = \arccos\left(\frac{\boldsymbol{a}_{inp}}{|\boldsymbol{a}_{inp}|} \cdot \boldsymbol{n}\right) \tag{4}$$

where $W_c^{\hat{y}}$ is the weight vector of the final (classifier) layer of the deep neural net (DNN) for the predicted class $\hat{y}$, $\boldsymbol{n}$ is the unit vector orthogonal to the decision boundary[1], $\boldsymbol{a}_{inp}$ is the latent space vector representation for the un-mixed input (i.e. activations of the penultimate layer of the DNN). Further, we ignore the bias term when calculating the angular margin therefore $W_c^{\hat{y}}$ does not include the bias term.

### 4.2 INTRA-CLASS MIXUP

In this subsection we explore the properties of intra-class mixup trained deep neural nets and how they can be leveraged to improve OoD detection performance. Inter-class mixup (Zhang et al., 2018) is a technique that helps improve the generalization of a machine learning classifier. Mixup trains a neural net on a convex combinations of examples and their corresponding soft labels. Consider a pair of training input, label tuples $(\boldsymbol{x}_i, y_i)$ and $(\boldsymbol{x}_j, y_j)$. Inter-class mixup obtains a new augmented sample $(\hat{\boldsymbol{x}}, \hat{y})$ for training the classifier described by Equation 5.

$$\begin{aligned} \hat{\boldsymbol{x}} \quad &= \lambda \cdot \boldsymbol{x}_i + (1 - \lambda) \cdot \boldsymbol{x}_j \\ \hat{y} \quad &= \lambda \cdot y_i + (1 - \lambda) \cdot y_j \end{aligned} \tag{5}$$

where, $\lambda \in [0, 1]$ represents the mixing parameter. In the same fashion as inter-class mixup (Zhang et al., 2018) $\lambda \sim \text{Beta}(\alpha, \alpha)$ and for our experiments we set $\alpha = 1$. Such an augmentation scheme forces the model to learn smooth interpolations between the samples of the training dataset. The proposed technique of intra-class mixup imposes an additional constraint on Equation 5 such that $\hat{y} = y_i = y_j$, forcing the training sample to be a convex combination of inputs within the same class. It can be shown that the additional constraint on $\hat{y}$ reduces the class conditional input variance (see Appendix A.1).

---

[1]The unit vector is the decision boundary normal in case of binary classification. In case of a multi-class classification, the interpretation depends on the training loss used.

$$\sigma_{\hat{X}}^2 = \left[\lambda^2 + (1 - \lambda)^2\right]\sigma_X^2$$

$$\text{Since } \lambda^2 + (1 - \lambda)^2 \le 1 \tag{6}$$

$$\sigma_{\hat{X}}^2 \le \sigma_X^2$$

where $\sigma_{\hat{X}}^2$ is the variance of the input after mixup and $\sigma_X^2$ is the variance of the input prior to mixup.

Therefore intra-class mixup results in reduction of class conditional variance at the input. We observe that deep neural nets trained on intra-class mixup learn to represent even unmixed images with lower angular spread (i.e. reduction of the class conditional representation variance) compared to ERM or inter-class mixup. The angular spread refers to the standard deviation of the angular margin for a given dataset. This reduced angular spread increases the angular margin between in-distribution and OoD samples improving OoD detection performance.

### 4.3 SUPPLEMENTING ANGULAR MARGIN

In this subsection we describe how angular margin scores can be supplemented with various OoD scores to improve detection performance. We observe from Figure 1, that traditional scores that use the $l_2$ margin can separate OoD data from in-distribution data. From Figure 1, this can be interpreted as higher values along the y-axis are more likely to be OoD. But using only $l_2$ ignores information in the angular margin that can be used to separate in-distribution samples from OoD samples. From Figure 1, we see that using angular margin improves separability (using two axes instead of one). This is also observed in our experimental results (refer to Table 3 and 4). Therefore we propose supplementing OoD score $J_s$ is given by

$$J_s = J + \cos(\theta) \tag{7}$$

where $J$ is the OoD score from a detector of choice, $\theta$ is the angular margin of the test sample described in Equation 4. The OoD score $J$ is different for different detection schemes. For MSP (Hendrycks & Gimpel, 2017) $J$ would be the softmax probability of the predicted class while $J$ would be the energy score (Liu et al., 2020) when using energy score detector, and so on. To complete our analysis we also use "Vanilla $\cos(\theta)$" which is when $J_s = \cos(\theta)$. The following section details various experiments to evaluate the performance of the proposed method.

## 5 EXPERIMENTS

We perform experiments which aim to understand and show the role of intra-class mixup in improving OoD detection performance. The experiments' goals are summarized below

- To show intra-class mixup trained networks learn representations that have lower angular spread and improved angular separability (Section 5.1).
- To show intra-class mixup improves OoD detection on methods that use distance proxies (Section 5.2).
- To show that supplemental use of angular margin improves OoD detection performance (Section 5.3).

### 5.1 SEPARABILITY

In this section we provide evidence for improved OoD separability resulting from *intra-class mixup*. To show that intra-class mixup improves OoD separability we evaluate the angular margin for in-distribution and OoD examples. The improvement in separability resulting from intra-class mixup can be observed using a separability metric $S$ given by

$$S = \frac{(\theta_o - \theta_i)^2}{\sigma_o^2 + \sigma_i^2} \tag{8}$$

where $\theta_o$ and $\theta_i$ are the angular margins for OoD data and in-distribution data respectively, $\sigma_o$ and $\sigma_i$ are the angular scatter/spread (i.e. standard deviation) of the angular margin. The metric $S$ is the angular version of the Fisher's criterion (Fisher et al., 1936).

| In-Dist | Average OoD Angular Margin (in rad) $\theta_o \pm \sigma_o$ | | | Average In-Dist. Angular Margin (in rad) $\theta_i \pm \sigma_i$ | | | Angular Separability $\frac{(\theta_o - \theta_i)^2}{\sigma_o^2 + \sigma_i^2}$ | | |
|---|---|---|---|---|---|---|---|---|---|
| | ERM | Inter | Intra | ERM | Inter | Intra | ERM | Inter | Intra |
| CIFAR-10 | 1.4262 ± 0.0587 | 1.2992 ± 0.0585 | 1.4534 ± 0.0478 | 1.2901 ± 0.0779 | 1.1977 ± 0.0480 | 1.3065 ± 0.0730 | 1.9469 | 1.7991 | **2.8342** |
| CIFAR-100 | 1.4481 ± 0.0474 | 1.2991 ± 0.0553 | 1.4819 ± 0.0426 | 1.3817 ± 0.0929 | 1.1942 ± 0.1017 | 1.4012 ± 0.0839 | 0.4053 | **0.8211** | 0.7355 |
| SVHN | 1.1810 ± 0.1174 | 1.2351 ± 0.0937 | 1.2626 ± 0.0952 | 0.8815 ± 0.1180 | 1.0501 ± 0.0596 | 0.9748 ± 0.1067 | 3.2375 | 2.7753 | **4.0507** |
| TinyImageNet | 1.1344 ± 0.1008 | 1.2990 ± 0.0444 | 1.2322 ± 0.0654 | 1.0931 ± 0.1623 | 1.2952 ± 0.0736 | 1.1955 ± 0.1146 | 0.0467 | 0.0020 | **0.0774** |

Table 1: Separability of intra-class mixup, inter-class mixup and ERM trained models (averaged over 5 different seeds and OoD datasets) on various datasets. Intra-class mixup has better separability in almost all cases.

Table 1 shows the angular separability of empirical risk minimization (ERM), inter-class mixup (Zhang et al., 2018) and the proposed intra-class mixup. From Table 1 we clearly see improved OoD separability from intra-class mixup. For each in-distribution dataset in Table 1, the numbers show the average angular margin and angular spread computed over 5 differently seeded ResNet18 networks evaluated on the test set of SVHN (Netzer et al., 2011), CIFAR-10, CIFAR-100 (Krizhevsky et al., 2009), Places365 (Zhou et al., 2017), Textures (Cimpoi et al., 2014), LSUN (Yu et al., 2015), TinyImageNet (Li et al.), Gaussian Noise and Uniform Noise as OoD datasets.

## 5.2 INTRA-CLASS MIXUP FOR OoD DETECTION

We have seen in the previous subsection that intra-class mixup improves OoD separability. In this subsection we provide evidence for improved OoD detection performance of intra-class mixup trained DNNs. We present the results of applying intra-class mixup on a few unsupervised methods that use proxies for distance from the decision boundary such as Maximum Softmax Probability (MSP) detection (Hendrycks & Gimpel, 2017), ODIN (Liang et al., 2018) and Energy score detection (Liu et al., 2020). To evaluate the performance of the detectors we use a variety of metrics which are briefly described in the following section.

**Metrics.** The performance of binary classification algorithm can be evaluated using Recall or True Positive Rate (TPR), False Positive Rate (FPR), Precision or a variety of other metrics. Recall (TPR), FPR and Precision are defined by equations (9), (10) and (11) respectively.

$$\text{Recall = True Positive Rate (TPR)} \quad = \frac{TP}{TP + FN} \tag{9}$$

$$\text{False Positive Rate (FPR)} \quad = \frac{FP}{FP + TN} \tag{10}$$

$$\text{Precision} \quad = \frac{TP}{TP + FP} \tag{11}$$

where, TP is True Positive, FP is False Positive, TN is True Negative and FN is False Negative. Metrics commonly used in literature (Hendrycks & Gimpel, 2017; Hendrycks et al., 2019; Lee et al., 2018b) to evaluate the performance of OoD detectors are:

| Dataset | Accuracy | | |
|---|---|---|---|
| | ERM | Inter | Intra |
| CIFAR-10 | $92.90 \pm 0.27$ | $94.32 \pm 0.29$ | $93.90 \pm 0.34$ |
| CIFAR-100 | $71.62 \pm 0.18$ | $76.51 \pm 0.28$ | $72.28 \pm 0.48$ |
| SVHN | $95.69 \pm 0.16$ | $96.42 \pm 0.04$ | $95.91 \pm 0.11$ |
| TinyImageNet | $32.80 \pm 0.17$ | $34.04 \pm 0.68$ | $34.63 \pm 0.30$ |

Table 2: Baseline accuracies (mean $\pm$ std. averaged over 5 seeds) of ERM, inter-class and intra-class mixup trained models on various datasets.

- **AUROC**: The Receiver Operating Characteristic (ROC), the plot of the True Positive Rate (TPR) against the False Positive Rate (FPR). The area under the ROC is called Area Under the Receiver Operating Characteristic (AUROC). AUROC of 1 denotes an ideal detection scheme, since the ideal detection algorithm results in 0 false positive and false negative samples.

- **AUPRC**: The Precision Recall Characteristic (PRC), the plot of the Precision against the Recall. The area under the PR is called AUPRC and should similarly be 1 for a ideal detection scheme.

- **FPR at TPR of 95% (FPR95)**: denotes the FPR when the TPR is 95%. Lower value of FPR at TPR of 95% indicates a better classifier.

To evaluate the effect of intra-class mixup on existing OoD detection techniques, we trained 5 differently seeded models each with ERM, inter-class mixup and intra-class mixup. We trained ResNet18 (He et al., 2016) networks on CIFAR-10, CIFAR-100, SVHN and TinyImageNet datasets till convergence (baseline accuracies are presented in Table 2). The training procedure used the SGD optimizer with a momentum of 0.9 and weight decay of $5 \times 10^{-4}$. The training used a 90%-10% training-validation split with the initial learning rate set to $10^{-2}$ and it was scaled down by a factor of 10 at 60% and 80% completion using a learning rate scheduler.

It is common practice in literature (Hendrycks & Gimpel, 2017; Lee et al., 2018b; Liang et al., 2018; Ren et al., 2019) to use data from distributions other than the training set as OoD data. We follow the same approach. For example, if a network is trained on CIFAR-10, datasets other than CIFAR-10 (SVHN, Places365, Textures, LSUN, TinyImageNet, Gaussian Noise and Uniform Noise) are used as OoD data. Table 3 (which is a condensed version of Table 5, 6 and 7) reports AUROC, AUPR and FPR95 results of applying MSP (Maximum Softmax Probability Detector (Hendrycks & Gimpel, 2017)), ODIN (Liang et al., 2018) and Energy Score (Liu et al., 2020) on ERM, inter-class mixup and intra-class mixup trained models. The detection performance (AUROC, AUPR and FPR95) was obtained by aggregating results from 5 differently seeded models and OoD datasets. We used Gaussian Noise, Uniform Noise, SVHN, CIFAR-10, CIFAR-100, Places365, Textures, LSUN and TinyImageNet as OoD datasets, excluding the corresponding in-distribution dataset.

We also compare the effect of using only $\cos(\theta)$ as the OoD score, referred in Table 3 as to "Vanilla $\cos(\theta)$". Using only $\cos(\theta)$ ignores information from the $l_2$ margin, thus when existing techniques are provided with angular information, we observe improved performance. Table 3 demonstrates that using "Vanilla $\cos(\theta)$" performs better than MSP, ODIN and Energy score based detectors. Further, in Table 4 we observe adding angular information to existing techniques enhances their performance.

On average we observe an improvement of **4.68%**, **3.98%** and **9.97%** on AUROC, AUPR and FPR95 metrics respectively on unsupervised OoD detection schemes with the use of intra-class mixup over empirical risk minimization. Further, we observe an improvement of **6.08%**, and **21.63%** on AUROC and FPR95 metrics over inter-class mixup and near identical AUPR performance when compared to inter-class mixup.

## 5.3 SUPPLEMENTING ANGULAR MARGIN

In this subsection we present the performance results when the OoD detection score is supplemented angular margin. We show that the use of the cosine of the angular margin as the OoD detection score

| Tech. | In-Dist. | AUROC ↑ | | | AUPR ↑ | | | FPR95 ↓ | | |
|---|---|---|---|---|---|---|---|---|---|---|
| | | ERM | Inter | Intra | ERM | Inter | Intra | ERM | Inter | Intra |
| MSP | CF-10 | 0.823 ± 0.054 | 0.884 ± 0.042 | **0.884 ± 0.037** | 0.780 ± 0.111 | **0.859 ± 0.084** | 0.819 ± 0.113 | 0.608 ± 0.111 | 0.504 ± 0.244 | **0.303 ± 0.096** |
| | CF-100 | 0.664 ± 0.168 | **0.773 ± 0.050** | 0.724 ± 0.020 | 0.620 ± 0.199 | **0.701 ± 0.162** | 0.651 ± 0.170 | 0.658 ± 0.130 | **0.555 ± 0.179** | 0.608 ± 0.109 |
| | SVHN | **0.923 ± 0.007** | 0.888 ± 0.062 | 0.900 ± 0.036 | 0.761 ± 0.109 | **0.783 ± 0.106** | 0.719 ± 0.126 | **0.266 ± 0.044** | 0.495 ± 0.197 | 0.338 ± 0.095 |
| | TIN | 0.551 ± 0.110 | 0.527 ± 0.083 | **0.581 ± 0.120** | 0.527 ± 0.199 | 0.514 ± 0.190 | **0.560 ± 0.206** | 0.826 ± 0.066 | 0.825 ± 0.050 | **0.809 ± 0.087** |
| | Avg. | 0.741 ± 0.177 | 0.768 ± 0.159 | **0.772 ± 0.146** | 0.672 ± 0.192 | **0.714 ± 0.191** | 0.688 ± 0.184 | 0.590 ± 0.224 | 0.595 ± 0.227 | **0.515 ± 0.228** |
| ODIN | CF-10 | 0.897 ± 0.017 | 0.772 ± 0.160 | **0.921 ± 0.031** | 0.852 ± 0.076 | 0.783 ± 0.150 | **0.878 ± 0.083** | 0.317 ± 0.086 | 0.594 ± 0.335 | **0.259 ± 0.116** |
| | CF-100 | 0.731 ± 0.090 | 0.763 ± 0.128 | **0.798 ± 0.065** | 0.656 ± 0.177 | **0.742 ± 0.202** | 0.721 ± 0.172 | 0.602 ± 0.156 | 0.621 ± 0.347 | **0.515 ± 0.216** |
| | SVHN | 0.836 ± 0.167 | **0.868 ± 0.080** | 0.855 ± 0.122 | 0.693 ± 0.216 | **0.806 ± 0.115** | 0.709 ± 0.188 | 0.518 ± 0.210 | 0.626 ± 0.115 | **0.480 ± 0.160** |
| | TIN | 0.642 ± 0.134 | 0.553 ± 0.041 | **0.680 ± 0.145** | 0.614 ± 0.220 | 0.532 ± 0.181 | **0.648 ± 0.242** | 0.738 ± 0.229 | 0.793 ± 0.092 | **0.680 ± 0.264** |
| | Avg. | 0.776 ± 0.152 | 0.739 ± 0.160 | **0.814 ± 0.135** | 0.704 ± 0.203 | 0.716 ± 0.198 | **0.739 ± 0.200** | 0.544 ± 0.236 | 0.658 ± 0.264 | **0.484 ± 0.248** |
| Energy Score | CF-10 | 0.880 ± 0.036 | 0.783 ± 0.164 | **0.903 ± 0.032** | 0.818 ± 0.096 | 0.812 ± 0.116 | **0.842 ± 0.087** | 0.322 ± 0.056 | 0.638 ± 0.344 | **0.276 ± 0.099** |
| | CF-100 | 0.672 ± 0.171 | 0.721 ± 0.096 | **0.748 ± 0.048** | 0.614 ± 0.202 | **0.669 ± 0.176** | 0.656 ± 0.177 | 0.637 ± 0.132 | 0.646 ± 0.261 | **0.554 ± 0.154** |
| | SVHN | **0.914 ± 0.014** | 0.881 ± 0.034 | 0.881 ± 0.035 | 0.764 ± 0.104 | **0.819 ± 0.085** | 0.703 ± 0.130 | **0.360 ± 0.072** | 0.611 ± 0.123 | 0.449 ± 0.100 |
| | TIN | 0.553 ± 0.097 | 0.533 ± 0.044 | **0.636 ± 0.135** | 0.523 ± 0.196 | 0.521 ± 0.179 | **0.597 ± 0.228** | 0.803 ± 0.084 | 0.801 ± 0.086 | **0.725 ± 0.180** |
| | Avg. | 0.753 ± 0.179 | 0.730 ± 0.163 | **0.792 ± 0.132** | 0.680 ± 0.197 | **0.705 ± 0.189** | 0.700 ± 0.187 | 0.530 ± 0.218 | 0.674 ± 0.240 | **0.501 ± 0.214** |
| Vanilla cos($\theta$) | CF-10 | 0.712 ± 0.130 | 0.873 ± 0.061 | **0.930 ± 0.022** | 0.836 ± 0.082 | 0.868 ± 0.070 | **0.882 ± 0.066** | 0.296 ± 0.083 | 0.512 ± 0.262 | **0.219 ± 0.117** |
| | CF-100 | 0.946 ± 0.007 | **0.796 ± 0.051** | 0.786 ± 0.051 | 0.638 ± 0.193 | **0.718 ± 0.163** | 0.694 ± 0.170 | 0.587 ± 0.146 | 0.501 ± 0.179 | **0.499 ± 0.194** |
| | SVHN | 0.939 ± 0.038 | 0.937 ± 0.040 | **0.958 ± 0.023** | 0.825 ± 0.084 | 0.854 ± 0.081 | **0.862 ± 0.080** | 0.196 ± 0.030 | 0.277 ± 0.138 | **0.152 ± 0.072** |
| | TIN | 0.564 ± 0.108 | 0.491 ± 0.101 | **0.583 ± 0.118** | 0.530 ± 0.198 | 0.489 ± 0.180 | **0.548 ± 0.202** | 0.796 ± 0.088 | 0.827 ± 0.073 | **0.773 ± 0.105** |
| | Avg. | 0.779 ± 0.174 | 0.774 ± 0.184 | **0.814 ± 0.163** | 0.707 ± 0.198 | 0.732 ± 0.202 | **0.747 ± 0.196** | 0.469 ± 0.256 | 0.529 ± 0.264 | **0.411 ± 0.278** |

Table 3: Various unsupervised distance proxy based OoD detection techniques on ERM, inter-class mixup and intra-class mixup trained models (averaged over 5 different seeds and OoD datasets) on various datasets (mean ± std). Expanded version of this table is available in Appendix A.2. CF-10 is CIFAR-10, CF-100 is CIFAR-100, TIN is TinyImageNet.

improves performance. The supplemented score is obtained by adding the OoD score to the cosine of the angular margin as described in Equation 7.

| Technique | In-Dataset | AUROC ↑ | | AUPR ↑ | | FPR95 ↓ | |
|---|---|---|---|---|---|---|---|
| | | Intra | Intra + Cos($\theta$) | Intra | Intra + Cos($\theta$) | Intra | Intra + Cos($\theta$) |
| MSP | CIFAR-10 | 0.884 ± 0.037 | **0.915 ± 0.020** | 0.819 ± 0.113 | **0.853 ± 0.089** | 0.303 ± 0.096 | **0.107 ± 0.030** |
| | CIFAR-100 | 0.724 ± 0.020 | **0.747 ± 0.020** | 0.651 ± 0.170 | **0.666 ± 0.168** | 0.608 ± 0.109 | **0.557 ± 0.127** |
| | SVHN | 0.900 ± 0.036 | **0.939 ± 0.023** | 0.719 ± 0.126 | **0.786 ± 0.105** | 0.338 ± 0.095 | **0.177 ± 0.063** |
| | TinyImageNet | 0.581 ± 0.120 | **0.583 ± 0.121** | **0.560 ± 0.206** | 0.558 ± 0.205 | 0.809 ± 0.087 | **0.799 ± 0.090** |
| | Average | 0.772 ± 0.146 | **0.796 ± 0.157** | 0.714 ± 0.191 | **0.716 ± 0.187** | 0.514 ± 0.228 | **0.441 ± 0.271** |
| ODIN | CIFAR-10 | 0.921 ± 0.031 | **0.935 ± 0.027** | 0.878 ± 0.083 | **0.900 ± 0.072** | 0.259 ± 0.116 | **0.229 ± 0.118** |
| | CIFAR-100 | 0.798 ± 0.065 | **0.828 ± 0.087** | 0.721 ± 0.172 | **0.764 ± 0.184** | 0.515 ± 0.216 | **0.462 ± 0.260** |
| | SVHN | 0.855 ± 0.122 | **0.933 ± 0.092** | 0.709 ± 0.188 | **0.840 ± 0.184** | 0.480 ± 0.160 | **0.221 ± 0.228** |
| | TinyImageNet | **0.680 ± 0.145** | 0.657 ± 0.150 | **0.648 ± 0.242** | 0.629 ± 0.237 | **0.680 ± 0.264** | 0.712 ± 0.221 |
| | Average | 0.814 ± 0.135 | **0.838 ± 0.150** | 0.739 ± 0.200 | **0.784 ± 0.206** | 0.483 ± 0.248 | **0.406 ± 0.293** |
| Energy Score | CIFAR-10 | 0.903 ± 0.032 | **0.919 ± 0.017** | 0.842 ± 0.087 | **0.863 ± 0.071** | 0.276 ± 0.099 | **0.238 ± 0.094** |
| | CIFAR-100 | 0.748 ± 0.048 | **0.763 ± 0.043** | 0.656 ± 0.177 | **0.700 ± 0.173** | 0.553 ± 0.154 | **0.532 ± 0.166** |
| | SVHN | 0.881 ± 0.035 | **0.927 ± 0.030** | 0.703 ± 0.130 | **0.798 ± 0.102** | 0.449 ± 0.100 | **0.279 ± 0.098** |
| | TinyImageNet | **0.635 ± 0.135** | 0.610 ± 0.125 | **0.597 ± 0.228** | 0.570 ± 0.214 | **0.725 ± 0.180** | 0.745 ± 0.143 |
| | Average | 0.792 ± 0.132 | **0.805 ± 0.147** | 0.700 ± 0.187 | **0.725 ± 0.189** | 0.501 ± 0.213 | **0.449 ± 0.242** |

Table 4: Various unsupervised distance proxy OoD detection techniques on compared with and without angular margin (averaged over 5 different seeds and OoD datasets, expanded version of this table is available in Appendix A.2) on various datasets (mean ± std).

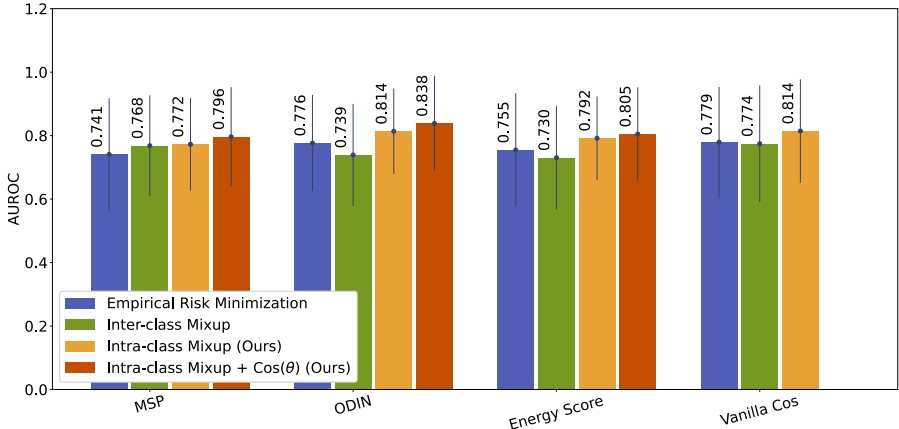

Figure 2: Comparison of average AUROC performance (averaged over in-dist and OoD datasets and 5 seeds) of methods that use distance proxies improved with the use of angular margin and intra-class mixup. Please note that "Vanilla cos" by definition cannot use the additional angular margin term, hence its not plotted.

Figure 2 and Table 4 compare the OoD detection performance of methods that use distance proxies when used with intra-class mixup aided with angular margin. Table 4 reports mean and standard deviation obtained by averaging 5 different seeds for each in-distribution dataset and averaged over various OoD datasets as used for Table 3. We observe that on average the use of angular margin with intra-class mixup improves OoD detection performance (AUROC) by **7.36%** over empirical risk minimization and **9.10%** over inter-class mixup.

## 6 CONCLUSION

There have been numerous approaches to tackle the problem of OoD detection. In this paper we explore the use of intra-class mixup to train OoD detectors. We show that intra-class mixup reduces the angular spread of the learnt latent space representation, improving the angular separability of in-distribution and out-of-distribution data. We observe that the use of intra-class mixup on detection techniques that use distance proxies improves AUROC performance by **4.68%** and **6.08%** over empirical risk minimization and inter-class mixup trained models respectively. Further, the use of the cosine of the angular margin to supplement detection scores improves AUROC performance by **7.36%** over empirical risk minimization and **9.10%** over inter-class mixup trained networks. Our findings reveal that intra-class mixup can be an effective tool for training OoD detectors.

## 7 IMPACT STATEMENT

The research presented in this paper focuses on improving the reliability of deep learning techniques in detecting anomalies, reducing the barriers to the deployment of deep learning in safety critical applications such as medical diagnosis, self driving cars etc. Hence it promises a positive impact on the society. However, one must be aware of the limitations of the specific techniques applied in conjunction with the technique proposed in the paper to ensure reliable and secure deployment in safety critical applications. This paper does not utilize human-derived data and does not utilize any datasets that have been discredited by the creators.

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

## A APPENDIX

### A.1 VARIANCE OF INPUT DUE TO INTRA-CLASS MIXUP

Consider two random variables $X_0, X_1 \sim X$ where $X$ is a class conditional distribution for class $\hat{y}$ with mean $\mu_X$ and variance $\sigma_X^2$. Then the variance of $\hat{X} = a_0 X_0 + a_1 X_1$ where $a_0 = \lambda$ and $a_1 = 1 - \lambda$ is given by

$$\sigma_{\hat{X}}^2 = Var(\hat{X}) = E\left[\left(\hat{X} - \mu_{\hat{X}}\right)^2\right]$$

$$\sigma_{\hat{X}}^2 = E\left[\left(\sum_{k=0}^1 a_i X_i - \sum_{k=0}^1 a_i \mu_X\right)^2\right]$$

$$\sigma_{\hat{X}}^2 = E\left[\left(\sum_{k=0}^1 a_k (X_k - \mu_X)\right)^2\right]$$

$$\sigma_{\hat{X}}^2 = E\left[\left(\sum_{k=0}^1 a_k (X_k - \mu_X)\right)\left(\sum_{k=0}^1 a_k (X_k - \mu_X)\right)\right]$$

$$\sigma_{\hat{X}}^2 = E\left[\sum_{j=0}^1 \sum_{k=0}^1 a_k a_j (X_k - \mu_X)(X_j - \mu_X)\right]$$

$$\sigma_{\hat{X}}^2 = \sum_{j=0}^1 \sum_{k=0}^1 a_k a_j E\left[(X_k - \mu_X)(X_j - \mu_X)\right]$$

$$\sigma_{\hat{X}}^2 = \sum_{k=0}^1 a_k^2 \sigma_X^2 + 2a_0 a_1 Cov(X_0, X_1)$$

The $Cov(X_0, X_1) = 0$ because $X_0$ and $X_1$ are drawn independently, therefore the equation is reduces to

$$\sigma_{\hat{X}}^2 = \left[\lambda^2 + (1 - \lambda)^2\right] \sigma_X^2$$

$$\text{Since } \lambda^2 + (1 - \lambda)^2 \leq 1$$

$$\sigma_{\hat{X}}^2 \leq \sigma_X^2$$

### A.2 EXPANDED RESULTS FOR VARIOUS DETECTION SCHEMES

| | | Maximum Softmax Probability Detector | | | | | | | | |
|---|---|---|---|---|---|---|---|---|---|---|
| | | AUROC | | | AUPR | | | FPR95 | | |
| In-Dataset | OoD Dataset | ERM | Inter | Intra | ERM | Inter | Intra | ERM | Inter | Intra |
| CIFAR-10 | Gaussian Noise | 0.6991 ± 0.2232 | 0.9470 ± 0.0471 | 0.8191 ± 0.0697 | 0.6451 ± 0.2250 | 0.9073 ± 0.0816 | 0.7118 ± 0.0975 | 0.6103 ± 0.3818 | 0.1290 ± 0.1128 | 0.3181 ± 0.0854 |
| | Uniform Noise | 0.8722 ± 0.0308 | 0.9462 ± 0.0383 | 0.9452 ± 0.0310 | 0.8005 ± 0.0413 | 0.9059 ± 0.0621 | 0.9009 ± 0.0691 | 0.3513 ± 0.2029 | 0.1383 ± 0.0960 | 0.1194 ± 0.0347 |
| | SVHN | 0.8111 ± 0.0901 | 0.8539 ± 0.0386 | 0.9175 ± 0.0105 | 0.9005 ± 0.0402 | 0.9263 ± 0.0167 | 0.9474 ± 0.0071 | 0.6102 ± 0.2288 | 0.6285 ± 0.1962 | 0.2135 ± 0.0230 |
| | Textures | 0.8448 ± 0.0089 | 0.8471 ± 0.0144 | 0.8674 ± 0.0064 | 0.7311 ± 0.0058 | 0.7677 ± 0.0147 | 0.7407 ± 0.0084 | 0.6698 ± 0.1196 | 0.7438 ± 0.0782 | 0.4066 ± 0.0330 |
| | LSUN | 0.8598 ± 0.0177 | 0.8888 ± 0.0063 | 0.8924 ± 0.0048 | 0.6262 ± 0.0203 | 0.7074 ± 0.0111 | 0.6447 ± 0.0069 | 0.6217 ± 0.1507 | 0.5010 ± 0.0439 | 0.3075 ± 0.0090 |
| | TinyImageNet | 0.8344 ± 0.0143 | 0.8484 ± 0.0046 | 0.8743 ± 0.0014 | 0.8169 ± 0.0078 | 0.8456 ± 0.0052 | 0.8409 ± 0.0023 | 0.7070 ± 0.1248 | 0.6929 ± 0.0187 | 0.3862 ± 0.0141 |
| | Places365 | 0.8422 ± 0.0155 | 0.8546 ± 0.0055 | 0.8758 ± 0.0033 | 0.9418 ± 0.0040 | 0.9510 ± 0.0018 | 0.9497 ± 0.0015 | 0.6851 ± 0.1368 | 0.6920 ± 0.0307 | 0.3745 ± 0.0045 |
| | Average | 0.8234 ± 0.0538 | 0.8837 ± 0.0419 | 0.8845 ± 0.0370 | 0.7803 ± 0.1113 | 0.8587 ± 0.0837 | 0.8194 ± 0.1126 | 0.6079 ± 0.1106 | 0.5037 ± 0.2445 | 0.3034 ± 0.0963 |
| CIFAR-100 | Gaussian Noise | 0.2596 ± 0.0665 | 0.8916 ± 0.0579 | 0.7189 ± 0.1093 | 0.3647 ± 0.0181 | 0.7963 ± 0.0875 | 0.6211 ± 0.1069 | 0.8237 ± 0.0798 | 0.2005 ± 0.0778 | 0.4668 ± 0.1020 |
| | Uniform Noise | 0.7941 ± 0.0627 | 0.7464 ± 0.2002 | 0.7159 ± 0.0908 | 0.6889 ± 0.0805 | 0.6705 ± 0.1818 | 0.6048 ± 0.0977 | 0.3759 ± 0.0618 | 0.3751 ± 0.2218 | 0.4384 ± 0.0742 |
| | SVHN | 0.7426 ± 0.0273 | 0.7460 ± 0.0302 | 0.7501 ± 0.0355 | 0.8507 ± 0.0173 | 0.8423 ± 0.0150 | 0.8560 ± 0.0211 | 0.6120 ± 0.0456 | 0.6137 ± 0.0796 | 0.5912 ± 0.0570 |
| | Textures | 0.6948 ± 0.0038 | 0.7454 ± 0.0030 | 0.6879 ± 0.0031 | 0.5075 ± 0.0046 | 0.5502 ± 0.0047 | 0.5031 ± 0.0046 | 0.7322 ± 0.0079 | 0.6451 ± 0.0133 | 0.7566 ± 0.0110 |
| | LSUN | 0.7020 ± 0.0051 | 0.7417 ± 0.0037 | 0.7138 ± 0.0018 | 0.3591 ± 0.0061 | 0.4014 ± 0.0076 | 0.3746 ± 0.0031 | 0.7094 ± 0.0059 | 0.7162 ± 0.0130 | 0.6931 ± 0.0059 |
| | TinyImageNet | 0.7382 ± 0.0026 | 0.7857 ± 0.0023 | 0.7494 ± 0.0032 | 0.6900 ± 0.0042 | 0.7447 ± 0.0022 | 0.7069 ± 0.0042 | 0.6518 ± 0.0025 | 0.6305 ± 0.0133 | 0.6393 ± 0.0037 |
| | Places365 | 0.7201 ± 0.0040 | 0.7560 ± 0.0019 | 0.7326 ± 0.0029 | 0.8818 ± 0.0022 | 0.8989 ± 0.0011 | 0.8885 ± 0.0018 | 0.6992 ± 0.0066 | 0.7035 ± 0.0043 | 0.6720 ± 0.0035 |
| | Average | 0.6645 ± 0.1680 | 0.7733 ± 0.0503 | 0.7241 ± 0.0203 | 0.6204 ± 0.1990 | 0.7006 ± 0.1619 | 0.6507 ± 0.1702 | 0.6577 ± 0.1304 | 0.5549 ± 0.1786 | 0.6083 ± 0.1090 |
| SVHN | Gaussian Noise | 0.9218 ± 0.0141 | 0.8198 ± 0.1368 | 0.8606 ± 0.0710 | 0.7770 ± 0.0298 | 0.6935 ± 0.2014 | 0.6570 ± 0.1413 | 0.2575 ± 0.0560 | 0.6185 ± 0.4020 | 0.4203 ± 0.1965 |
| | Uniform Noise | 0.9247 ± 0.0061 | 0.7706 ± 0.1228 | 0.8336 ± 0.0760 | 0.7845 ± 0.0061 | 0.6385 ± 0.1586 | 0.6170 ± 0.1391 | 0.2494 ± 0.0335 | 0.7598 ± 0.3313 | 0.4974 ± 0.1963 |
| | CIFAR-100 | 0.9267 ± 0.0042 | 0.9340 ± 0.0071 | 0.9285 ± 0.0031 | 0.7993 ± 0.0067 | 0.8672 ± 0.0108 | 0.8072 ± 0.0085 | 0.2552 ± 0.0257 | 0.3282 ± 0.0841 | 0.2549 ± 0.0112 |
| | Textures | 0.9066 ± 0.0059 | 0.8851 ± 0.0253 | 0.9019 ± 0.0142 | 0.6651 ± 0.0154 | 0.7391 ± 0.0417 | 0.6643 ± 0.0299 | 0.3723 ± 0.0364 | 0.7703 ± 0.1736 | 0.4037 ± 0.0827 |
| | LSUN | 0.9258 ± 0.0051 | 0.9353 ± 0.0110 | 0.9176 ± 0.0063 | 0.5614 ± 0.0133 | 0.7183 ± 0.0313 | 0.5419 ± 0.0200 | 0.2534 ± 0.0260 | 0.3332 ± 0.1233 | 0.2964 ± 0.0263 |
| | TinyImageNet | 0.9308 ± 0.0037 | 0.9354 ± 0.0105 | 0.9339 ± 0.0035 | 0.8071 ± 0.0066 | 0.8732 ± 0.0176 | 0.8160 ± 0.0091 | 0.2306 ± 0.0221 | 0.3254 ± 0.1094 | 0.2300 ± 0.0146 |
| | Places365 | 0.9281 ± 0.0047 | 0.9355 ± 0.0133 | 0.9266 ± 0.0052 | 0.9324 ± 0.0034 | 0.9540 ± 0.0087 | 0.9325 ± 0.0047 | 0.2436 ± 0.0256 | 0.3307 ± 0.1388 | 0.2607 ± 0.0220 |
| | Average | 0.9235 ± 0.0074 | 0.8880 ± 0.0624 | 0.9003 ± 0.0360 | 0.7610 ± 0.1088 | 0.7834 ± 0.1065 | 0.7194 ± 0.1261 | 0.2660 ± 0.0442 | 0.4952 ± 0.1967 | 0.3377 ± 0.0948 |
| TinyImageNet | Gaussian Noise | 0.5506 ± 0.1104 | 0.4041 ± 0.1878 | 0.7082 ± 0.1253 | 0.4911 ± 0.0665 | 0.4355 ± 0.1123 | 0.6584 ± 0.1311 | 0.7374 ± 0.1040 | 0.7842 ± 0.1274 | 0.6326 ± 0.1357 |
| | Uniform Noise | 0.2977 ± 0.0686 | 0.3944 ± 0.1244 | 0.3135 ± 0.1295 | 0.3726 ± 0.0220 | 0.4104 ± 0.0488 | 0.3842 ± 0.0420 | 0.9064 ± 0.0255 | 0.7805 ± 0.1021 | 0.8873 ± 0.0470 |
| | SVHN | 0.6747 ± 0.0258 | 0.6042 ± 0.0915 | 0.6870 ± 0.0422 | 0.7956 ± 0.0237 | 0.7473 ± 0.0607 | 0.8233 ± 0.0262 | 0.7187 ± 0.0201 | 0.7458 ± 0.0843 | 0.7290 ± 0.0445 |
| | CIFAR-100 | 0.5774 ± 0.0034 | 0.5680 ± 0.0098 | 0.5934 ± 0.0033 | 0.5491 ± 0.0038 | 0.5423 ± 0.0095 | 0.5644 ± 0.0040 | 0.8718 ± 0.0035 | 0.8819 ± 0.0034 | 0.8622 ± 0.0031 |
| | Textures | 0.5555 ± 0.0078 | 0.5435 ± 0.0143 | 0.5610 ± 0.0109 | 0.3801 ± 0.0099 | 0.3756 ± 0.0091 | 0.3913 ± 0.0093 | 0.8702 ± 0.0056 | 0.8786 ± 0.0094 | 0.8809 ± 0.0058 |
| | LSUN | 0.5897 ± 0.0051 | 0.5727 ± 0.0263 | 0.5892 ± 0.0047 | 0.2709 ± 0.0041 | 0.2615 ± 0.0186 | 0.2690 ± 0.0044 | 0.8451 ± 0.0044 | 0.8639 ± 0.0185 | 0.8426 ± 0.0057 |
| | Places365 | 0.6117 ± 0.0045 | 0.6032 ± 0.0106 | 0.6137 ± 0.0029 | 0.8309 ± 0.0014 | 0.8270 ± 0.0061 | 0.8316 ± 0.0013 | 0.8334 ± 0.0042 | 0.8424 ± 0.0070 | 0.8271 ± 0.0025 |
| | Average | 0.5510 ± 0.1104 | 0.5272 ± 0.0832 | 0.5808 ± 0.1199 | 0.5272 ± 0.1990 | 0.5142 ± 0.1901 | 0.5603 ± 0.2059 | 0.8261 ± 0.0658 | 0.8253 ± 0.0505 | 0.8088 ± 0.0871 |
| Average | | 0.7406 ± 0.1771 | 0.7680 ± 0.1588 | 0.7725 ± 0.1461 | 0.6722 ± 0.1915 | 0.7142 ± 0.1914 | 0.6875 ± 0.1843 | 0.5895 ± 0.2243 | 0.5948 ± 0.2269 | 0.5145 ± 0.2285 |

Table 5: Expanded results for Maximum Softmax Probability Detector.

| | | ODIN | | | | | | | | |
|---|---|---|---|---|---|---|---|---|---|---|
| | | AUROC | | | AUPR | | | FPR95 | | |
| In-Dataset | OoD Dataset | ERM | Inter | Intra | ERM | Inter | Intra | ERM | Inter | Intra |
| CIFAR-10 | Gaussian Noise | 0.9059 ± 0.0866 | 0.9942 ± 0.0109 | 0.8898 ± 0.0665 | 0.8378 ± 0.1369 | 0.9935 ± 0.0121 | 0.8038 ± 0.1183 | 0.1691 ± 0.1213 | 0.0282 ± 0.0544 | 0.2091 ± 0.1046 |
| | Uniform Noise | 0.8624 ± 0.0585 | 0.9576 ± 0.0370 | 0.9738 ± 0.0105 | 0.7709 ± 0.0863 | 0.9335 ± 0.0620 | 0.9453 ± 0.0287 | 0.2807 ± 0.1075 | 0.1214 ± 0.0857 | 0.0581 ± 0.0173 |
| | SVHN | 0.9127 ± 0.0441 | 0.4742 ± 0.0900 | 0.9598 ± 0.0116 | 0.9490 ± 0.0260 | 0.7191 ± 0.0595 | 0.9801 ± 0.0057 | 0.2489 ± 0.1176 | 0.9100 ± 0.0368 | 0.1498 ± 0.0388 |
| | Textures | 0.8993 ± 0.0202 | 0.7156 ± 0.0405 | 0.8927 ± 0.0128 | 0.8237 ± 0.0266 | 0.6403 ± 0.0381 | 0.8078 ± 0.0188 | 0.3650 ± 0.0836 | 0.8312 ± 0.0517 | 0.3799 ± 0.0437 |
| | LSUN | 0.9151 ± 0.0049 | 0.7959 ± 0.0193 | 0.9243 ± 0.0051 | 0.7474 ± 0.0074 | 0.5633 ± 0.0372 | 0.7552 ± 0.0123 | 0.3184 ± 0.0258 | 0.6734 ± 0.0249 | 0.2745 ± 0.0233 |
| | TinyImageNet | 0.8840 ± 0.0130 | 0.7257 ± 0.0129 | 0.9010 ± 0.0068 | 0.8696 ± 0.0123 | 0.7264 ± 0.0137 | 0.8867 ± 0.0074 | 0.4406 ± 0.0560 | 0.8021 ± 0.0123 | 0.3857 ± 0.0302 |
| | Places365 | 0.8987 ± 0.0081 | 0.7429 ± 0.0175 | 0.9086 ± 0.0004 | 0.9642 ± 0.0028 | 0.9085 ± 0.0078 | 0.9672 ± 0.0004 | 0.3942 ± 0.0390 | 0.7878 ± 0.0188 | 0.3533 ± 0.0036 |
| | Average | 0.8969 ± 0.0170 | 0.7723 ± 0.1600 | 0.9214 ± 0.0308 | 0.8518 ± 0.0764 | 0.7835 ± 0.1505 | 0.8780 ± 0.0832 | 0.3167 ± 0.0857 | 0.5935 ± 0.3352 | 0.2587 ± 0.1162 |
| CIFAR-100 | Gaussian Noise | 0.5267 ± 0.0759 | 0.9988 ± 0.0015 | 0.9365 ± 0.0374 | 0.4631 ± 0.0373 | 0.9979 ± 0.0026 | 0.8727 ± 0.0757 | 0.5671 ± 0.0788 | 0.0038 ± 0.0051 | 0.1305 ± 0.0661 |
| | Uniform Noise | 0.8447 ± 0.0539 | 0.9187 ± 0.1022 | 0.8070 ± 0.1008 | 0.7173 ± 0.0776 | 0.8686 ± 0.1521 | 0.6811 ± 0.1122 | 0.2490 ± 0.0735 | 0.1534 ± 0.1610 | 0.2741 ± 0.1211 |
| | SVHN | 0.7614 ± 0.0249 | 0.7258 ± 0.0377 | 0.8147 ± 0.0247 | 0.8480 ± 0.0182 | 0.8599 ± 0.0256 | 0.8859 ± 0.0201 | 0.5813 ± 0.0369 | 0.7577 ± 0.0412 | 0.4685 ± 0.0363 |
| | Textures | 0.7347 ± 0.0060 | 0.6878 ± 0.0074 | 0.7080 ± 0.0094 | 0.5533 ± 0.0068 | 0.5830 ± 0.0066 | 0.5214 ± 0.0095 | 0.7024 ± 0.0105 | 0.8855 ± 0.0086 | 0.7478 ± 0.0202 |
| | LSUN | 0.7323 ± 0.0069 | 0.6783 ± 0.0065 | 0.7625 ± 0.0024 | 0.3957 ± 0.0093 | 0.3754 ± 0.0095 | 0.4376 ± 0.0036 | 0.7386 ± 0.0127 | 0.8414 ± 0.0045 | 0.6831 ± 0.0113 |
| | TinyImageNet | 0.7635 ± 0.0033 | 0.6510 ± 0.0059 | 0.7790 ± 0.0012 | 0.7197 ± 0.0051 | 0.6329 ± 0.0062 | 0.7376 ± 0.0016 | 0.6707 ± 0.0064 | 0.8698 ± 0.0032 | 0.6370 ± 0.0050 |
| | Places365 | 0.7515 ± 0.0062 | 0.6780 ± 0.0025 | 0.7778 ± 0.0028 | 0.8972 ± 0.0032 | 0.8735 ± 0.0013 | 0.9104 ± 0.0015 | 0.7064 ± 0.0097 | 0.8328 ± 0.0074 | 0.6624 ± 0.0059 |
| | Average | 0.7307 ± 0.0903 | 0.7626 ± 0.1275 | 0.7979 ± 0.0651 | 0.6563 ± 0.1766 | 0.7416 ± 0.2035 | 0.7210 ± 0.1724 | 0.6022 ± 0.1561 | 0.6206 ± 0.3472 | 0.5148 ± 0.2162 |
| SVHN | Gaussian Noise | 0.8929 ± 0.0257 | 0.8318 ± 0.1540 | 0.8902 ± 0.0627 | 0.7453 ± 0.0509 | 0.7685 ± 0.1972 | 0.7592 ± 0.1302 | 0.4429 ± 0.1016 | 0.5501 ± 0.3915 | 0.4052 ± 0.2158 |
| | Uniform Noise | 0.4328 ± 0.0662 | 0.6939 ± 0.1668 | 0.5604 ± 0.1762 | 0.2320 ± 0.0232 | 0.5804 ± 0.1992 | 0.3311 ± 0.1563 | 0.9430 ± 0.0224 | 0.7344 ± 0.3006 | 0.7954 ± 0.2306 |
| | CIFAR-100 | 0.9216 ± 0.0105 | 0.9217 ± 0.0129 | 0.9264 ± 0.0098 | 0.8360 ± 0.0145 | 0.8877 ± 0.0149 | 0.8444 ± 0.0178 | 0.3803 ± 0.0717 | 0.5923 ± 0.0875 | 0.3507 ± 0.0491 |
| | Textures | 0.8375 ± 0.0083 | 0.8494 ± 0.0382 | 0.8539 ± 0.0330 | 0.6088 ± 0.0133 | 0.7524 ± 0.0522 | 0.6394 ± 0.0548 | 0.7294 ± 0.0526 | 0.8615 ± 0.0566 | 0.6461 ± 0.1253 |
| | LSUN | 0.9216 ± 0.0122 | 0.9283 ± 0.0171 | 0.9081 ± 0.0180 | 0.6500 ± 0.0331 | 0.8052 ± 0.0318 | 0.6110 ± 0.0497 | 0.3831 ± 0.0715 | 0.5483 ± 0.1222 | 0.4339 ± 0.0754 |
| | TinyImageNet | 0.9251 ± 0.0091 | 0.9262 ± 0.0173 | 0.9276 ± 0.0117 | 0.8422 ± 0.0142 | 0.8953 ± 0.0210 | 0.8451 ± 0.0213 | 0.3628 ± 0.0570 | 0.5571 ± 0.1221 | 0.3414 ± 0.0593 |
| | Places365 | 0.9201 ± 0.0111 | 0.9255 ± 0.0212 | 0.9168 ± 0.0160 | 0.9386 ± 0.0076 | 0.9559 ± 0.0121 | 0.9355 ± 0.0121 | 0.3843 ± 0.0656 | 0.5415 ± 0.1555 | 0.3873 ± 0.0763 |
| | Average | 0.8359 ± 0.1671 | 0.8681 ± 0.0803 | 0.8548 ± 0.1225 | 0.6933 ± 0.2162 | 0.8065 ± 0.1148 | 0.7094 ± 0.1882 | 0.5180 ± 0.2105 | 0.6265 ± 0.1147 | 0.4800 ± 0.1601 |
| TinyImageNet | Gaussian Noise | 0.9353 ± 0.0691 | 0.5092 ± 0.2553 | 0.9772 ± 0.0219 | 0.9130 ± 0.0960 | 0.5169 ± 0.2044 | 0.9737 ± 0.0263 | 0.1958 ± 0.1672 | 0.6574 ± 0.2351 | 0.0943 ± 0.0854 |
| | Uniform Noise | 0.4737 ± 0.0700 | 0.4829 ± 0.1432 | 0.5562 ± 0.1663 | 0.4421 ± 0.0348 | 0.4541 ± 0.0796 | 0.5037 ± 0.0968 | 0.7836 ± 0.0461 | 0.6827 ± 0.1446 | 0.6776 ± 0.1424 |
| | SVHN | 0.6754 ± 0.0380 | 0.5840 ± 0.1369 | 0.8180 ± 0.0125 | 0.8018 ± 0.0315 | 0.7332 ± 0.0748 | 0.9070 ± 0.0054 | 0.7237 ± 0.0281 | 0.7344 ± 0.1257 | 0.5611 ± 0.0282 |
| | CIFAR-100 | 0.6016 ± 0.0034 | 0.5700 ± 0.0141 | 0.6162 ± 0.0032 | 0.5805 ± 0.0037 | 0.5461 ± 0.0124 | 0.5940 ± 0.0032 | 0.8754 ± 0.0051 | 0.8861 ± 0.0128 | 0.8576 ± 0.0048 |
| | Textures | 0.5576 ± 0.0067 | 0.5436 ± 0.0295 | 0.5700 ± 0.0098 | 0.4226 ± 0.0084 | 0.3808 ± 0.0177 | 0.4415 ± 0.0132 | 0.9145 ± 0.0032 | 0.8840 ± 0.0391 | 0.9140 ± 0.0050 |
| | LSUN | 0.6252 ± 0.0052 | 0.5714 ± 0.0381 | 0.6045 ± 0.0052 | 0.3022 ± 0.0056 | 0.2641 ± 0.0298 | 0.2803 ± 0.0051 | 0.8381 ± 0.0056 | 0.8672 ± 0.0258 | 0.8350 ± 0.0061 |
| | Places365 | 0.6251 ± 0.0033 | 0.6077 ± 0.0181 | 0.6216 ± 0.0030 | 0.8388 ± 0.0020 | 0.8297 ± 0.0089 | 0.8344 ± 0.0016 | 0.8373 ± 0.0046 | 0.8413 ± 0.0241 | 0.8232 ± 0.0041 |
| | Average | 0.6420 ± 0.1335 | 0.5527 ± 0.0406 | 0.6805 ± 0.1454 | 0.6144 ± 0.2203 | 0.5321 ± 0.1814 | 0.6478 ± 0.2419 | 0.7383 ± 0.2287 | 0.7933 ± 0.0916 | 0.6804 ± 0.2641 |
| Average | | 0.7764 ± 0.1520 | 0.7389 ± 0.1605 | 0.8137 ± 0.1347 | 0.7040 ± 0.2028 | 0.7159 ± 0.1978 | 0.7390 ± 0.1996 | 0.5438 ± 0.2355 | 0.6585 ± 0.2642 | 0.4835 ± 0.2479 |

Table 6: Expanded results for ODIN.

| | | AUROC | | | AUPR | | | FPR95 | | |
|---|---|---|---|---|---|---|---|---|---|---|
| In-Dataset | OoD Dataset | ERM | Inter | Intra | ERM | Inter | Intra | ERM | Inter | Intra |
| CIFAR-10 | Gaussian Noise | 0.7997 ± 0.1342 | 0.9694 ± 0.0422 | 0.8390 ± 0.0910 | 0.6885 ± 0.1384 | 0.9496 ± 0.0658 | 0.7318 ± 0.1090 | 0.2975 ± 0.1625 | 0.0966 ± 0.1382 | 0.2829 ± 0.1262 |
| | Uniform Noise | 0.8719 ± 0.0356 | 0.9645 ± 0.0197 | 0.9249 ± 0.0402 | 0.7576 ± 0.0457 | 0.9294 ± 0.0330 | 0.8322 ± 0.0862 | 0.2446 ± 0.0688 | 0.1060 ± 0.0700 | 0.1192 ± 0.0544 |
| | SVHN | 0.8880 ± 0.0354 | 0.4423 ± 0.0759 | 0.9266 ± 0.0111 | 0.9268 ± 0.0191 | 0.7171 ± 0.0385 | 0.9422 ± 0.0088 | 0.2773 ± 0.0891 | 0.9542 ± 0.0316 | 0.1839 ± 0.0294 |
| | Textures | 0.8849 ± 0.0114 | 0.7291 ± 0.0382 | 0.8742 ± 0.0099 | 0.7669 ± 0.0132 | 0.6743 ± 0.0357 | 0.7548 ± 0.0152 | 0.3722 ± 0.0541 | 0.8883 ± 0.0343 | 0.4320 ± 0.0437 |
| | LSUN | 0.9201 ± 0.0041 | 0.8262 ± 0.0181 | 0.9346 ± 0.0052 | 0.7494 ± 0.0104 | 0.6769 ± 0.0252 | 0.7747 ± 0.0169 | 0.2860 ± 0.0119 | 0.7493 ± 0.0224 | 0.2343 ± 0.0136 |
| | TinyImageNet | 0.8899 ± 0.0072 | 0.7721 ± 0.0096 | 0.9057 ± 0.0044 | 0.8724 ± 0.0071 | 0.8021 ± 0.0087 | 0.8898 ± 0.0049 | 0.4086 ± 0.0326 | 0.8286 ± 0.0081 | 0.3584 ± 0.0197 |
| | Places365 | 0.9030 ± 0.0059 | 0.7793 ± 0.0128 | 0.9156 ± 0.0040 | 0.9650 ± 0.0021 | 0.9318 ± 0.0042 | 0.9693 ± 0.0016 | 0.3693 ± 0.0225 | 0.8419 ± 0.0152 | 0.3225 ± 0.0126 |
| | Average | 0.8796 ± 0.0355 | 0.7833 ± 0.1642 | 0.9029 ± 0.0319 | 0.8181 ± 0.0957 | 0.8116 ± 0.1155 | 0.8421 ± 0.0870 | 0.3222 ± 0.0562 | 0.6379 ± 0.3441 | 0.2762 ± 0.0987 |
| CIFAR-100 | Gaussian Noise | 0.2613 ± 0.0566 | 0.9158 ± 0.0781 | 0.8128 ± 0.0815 | 0.3657 ± 0.0169 | 0.8444 ± 0.1359 | 0.6856 ± 0.0920 | 0.8050 ± 0.0531 | 0.1430 ± 0.1170 | 0.2883 ± 0.0914 |
| | Uniform Noise | 0.7004 ± 0.0573 | 0.7754 ± 0.2019 | 0.6778 ± 0.0991 | 0.5664 ± 0.0464 | 0.7035 ± 0.1933 | 0.5565 ± 0.0740 | 0.3918 ± 0.0625 | 0.3473 ± 0.2581 | 0.4136 ± 0.1134 |
| | SVHN | 0.7763 ± 0.0206 | 0.5954 ± 0.0330 | 0.7686 ± 0.0230 | 0.8414 ± 0.0158 | 0.7434 ± 0.0234 | 0.8269 ± 0.0137 | 0.5026 ± 0.0291 | 0.8058 ± 0.0457 | 0.4757 ± 0.0466 |
| | Textures | 0.7045 ± 0.0072 | 0.6451 ± 0.0029 | 0.6860 ± 0.0087 | 0.5062 ± 0.0065 | 0.4727 ± 0.0065 | 0.4850 ± 0.0068 | 0.7266 ± 0.0132 | 0.8539 ± 0.0122 | 0.7586 ± 0.0162 |
| | LSUN | 0.7195 ± 0.0057 | 0.6886 ± 0.0082 | 0.7295 ± 0.0038 | 0.3714 ± 0.0073 | 0.3550 ± 0.0104 | 0.3775 ± 0.0054 | 0.7125 ± 0.0101 | 0.8057 ± 0.0094 | 0.6662 ± 0.0075 |
| | TinyImageNet | 0.7855 ± 0.0024 | 0.7350 ± 0.0047 | 0.7942 ± 0.0010 | 0.7478 ± 0.0033 | 0.6916 ± 0.0071 | 0.7594 ± 0.0027 | 0.6305 ± 0.0018 | 0.7495 ± 0.0058 | 0.6150 ± 0.0039 |
| | Places365 | 0.7569 ± 0.0040 | 0.6909 ± 0.0006 | 0.7665 ± 0.0041 | 0.9009 ± 0.0020 | 0.8720 ± 0.0008 | 0.9043 ± 0.0025 | 0.6873 ± 0.0055 | 0.8165 ± 0.0103 | 0.6574 ± 0.0082 |
| | Average | 0.6721 ± 0.1706 | 0.7209 ± 0.0961 | 0.7479 ± 0.0481 | 0.6143 ± 0.2022 | 0.6689 ± 0.1758 | 0.6565 ± 0.1769 | 0.6366 ± 0.1324 | 0.6460 ± 0.2609 | 0.5535 ± 0.1537 |
| SVHN | Gaussian Noise | 0.9150 ± 0.0220 | 0.8192 ± 0.1718 | 0.8406 ± 0.0894 | 0.7778 ± 0.0484 | 0.7466 ± 0.2167 | 0.6500 ± 0.1680 | 0.3248 ± 0.0859 | 0.5670 ± 0.4085 | 0.4952 ± 0.2378 |
| | Uniform Noise | 0.9007 ± 0.0148 | 0.7946 ± 0.1221 | 0.8251 ± 0.0875 | 0.7371 ± 0.0478 | 0.7089 ± 0.1539 | 0.6260 ± 0.1636 | 0.3753 ± 0.0504 | 0.6984 ± 0.3347 | 0.5352 ± 0.2273 |
| | CIFAR-100 | 0.9229 ± 0.0072 | 0.9218 ± 0.0121 | 0.9122 ± 0.0101 | 0.8174 ± 0.0136 | 0.8807 ± 0.0148 | 0.7858 ± 0.0208 | 0.3321 ± 0.0450 | 0.5889 ± 0.0847 | 0.3605 ± 0.0459 |
| | Textures | 0.8862 ± 0.0080 | 0.8616 ± 0.0343 | 0.8639 ± 0.0245 | 0.6555 ± 0.0267 | 0.7611 ± 0.0505 | 0.6281 ± 0.0447 | 0.5292 ± 0.0311 | 0.8726 ± 0.0616 | 0.6217 ± 0.1184 |
| | LSUN | 0.9203 ± 0.0100 | 0.9293 ± 0.0170 | 0.8928 ± 0.0210 | 0.5992 ± 0.0367 | 0.7856 ± 0.0333 | 0.5053 ± 0.0525 | 0.3448 ± 0.0522 | 0.5317 ± 0.1322 | 0.4416 ± 0.0824 |
| | TinyImageNet | 0.9283 ± 0.0067 | 0.9308 ± 0.0147 | 0.9197 ± 0.0117 | 0.8266 ± 0.0155 | 0.8942 ± 0.0196 | 0.7994 ± 0.0244 | 0.2979 ± 0.0351 | 0.5233 ± 0.1276 | 0.3210 ± 0.0475 |
| | Places365 | 0.9243 ± 0.0086 | 0.9309 ± 0.0196 | 0.9111 ± 0.0164 | 0.9363 ± 0.0076 | 0.9574 ± 0.0114 | 0.9235 ± 0.0134 | 0.3179 ± 0.0443 | 0.4968 ± 0.1675 | 0.3668 ± 0.0740 |
| | Average | 0.9140 ± 0.0140 | 0.8840 ± 0.0542 | 0.8808 ± 0.0350 | 0.7643 ± 0.1044 | 0.8192 ± 0.0849 | 0.7026 ± 0.1300 | 0.3603 ± 0.0724 | 0.6113 ± 0.1228 | 0.4489 ± 0.1003 |
| TinyImageNet | Gaussian Noise | 0.5227 ± 0.1378 | 0.4536 ± 0.2639 | 0.8491 ± 0.1009 | 0.4717 ± 0.0648 | 0.4843 ± 0.1818 | 0.7936 ± 0.1324 | 0.6961 ± 0.1173 | 0.6887 ± 0.2240 | 0.3730 ± 0.1908 |
| | Uniform Noise | 0.3399 ± 0.0730 | 0.4879 ± 0.1808 | 0.4347 ± 0.1476 | 0.3868 ± 0.0259 | 0.4730 ± 0.1057 | 0.4313 ± 0.0622 | 0.8750 ± 0.0345 | 0.6881 ± 0.1671 | 0.7789 ± 0.1042 |
| | SVHN | 0.6734 ± 0.0499 | 0.5505 ± 0.1357 | 0.8127 ± 0.0385 | 0.7781 ± 0.0410 | 0.7066 ± 0.0674 | 0.8950 ± 0.0267 | 0.6508 ± 0.0418 | 0.7373 ± 0.1069 | 0.5301 ± 0.0585 |
| | CIFAR-100 | 0.5778 ± 0.0041 | 0.5522 ± 0.0167 | 0.5958 ± 0.0032 | 0.5500 ± 0.0031 | 0.5287 ± 0.0136 | 0.5691 ± 0.0028 | 0.8775 ± 0.0042 | 0.8945 ± 0.0112 | 0.8663 ± 0.0029 |
| | Textures | 0.5620 ± 0.0187 | 0.5396 ± 0.0378 | 0.5874 ± 0.0114 | 0.3813 ± 0.0185 | 0.3839 ± 0.0279 | 0.4200 ± 0.0131 | 0.8539 ± 0.0055 | 0.8886 ± 0.0234 | 0.8650 ± 0.0120 |
| | LSUN | 0.5810 ± 0.0081 | 0.5489 ± 0.0454 | 0.5598 ± 0.0069 | 0.2612 ± 0.0063 | 0.2487 ± 0.0306 | 0.2461 ± 0.0048 | 0.8436 ± 0.0051 | 0.8699 ± 0.0274 | 0.8457 ± 0.0085 |
| | Places365 | 0.6135 ± 0.0038 | 0.5972 ± 0.0276 | 0.6087 ± 0.0037 | 0.8287 ± 0.0019 | 0.8241 ± 0.0129 | 0.8265 ± 0.0016 | 0.8212 ± 0.0038 | 0.8411 ± 0.0292 | 0.8189 ± 0.0045 |
| | Average | 0.5529 ± 0.0971 | 0.5328 ± 0.0438 | 0.6355 ± 0.1350 | 0.5226 ± 0.1961 | 0.5213 ± 0.1785 | 0.5974 ± 0.2277 | 0.8026 ± 0.0845 | 0.8012 ± 0.0863 | 0.7254 ± 0.1804 |
| Average | | 0.7546 ± 0.1792 | 0.7303 ± 0.1632 | 0.7918 ± 0.1318 | 0.6798 ± 0.1966 | 0.7053 ± 0.1889 | 0.6996 ± 0.1872 | 0.5304 ± 0.2184 | 0.6741 ± 0.2404 | 0.5010 ± 0.2135 |

Table 7: Expanded results for Energy Score.

| | | AUROC | | AUPR | | FPR95 | |
|---|---|---|---|---|---|---|---|
| In-Dataset | OoD Dataset | Intra | Intra + Cos($\theta$) | Intra | Intra + Cos($\theta$) | Intra | Intra + Cos($\theta$) |
| | | | | Maximum Softmax Probability Detector | | | |
| CIFAR-10 | Gaussian Noise | $0.8191 \pm 0.0697$ | $0.9207 \pm 0.0195$ | $0.7116 \pm 0.0969$ | $0.8191 \pm 0.0463$ | $0.3168 \pm 0.0862$ | $0.1070 \pm 0.0294$ |
| | Uniform Noise | $0.9452 \pm 0.0310$ | $0.9549 \pm 0.0229$ | $0.9007 \pm 0.0698$ | $0.9105 \pm 0.0599$ | $0.1188 \pm 0.0341$ | $0.0894 \pm 0.0186$ |
| | SVHN | $0.9175 \pm 0.0105$ | $0.9209 \pm 0.0101$ | $0.9474 \pm 0.0071$ | $0.9486 \pm 0.0068$ | $0.2135 \pm 0.0230$ | $0.2023 \pm 0.0344$ |
| | Textures | $0.8674 \pm 0.0064$ | $0.9049 \pm 0.0098$ | $0.7407 \pm 0.0084$ | $0.7806 \pm 0.0116$ | $0.4066 \pm 0.0330$ | $0.2613 \pm 0.0472$ |
| | LSUN | $0.8924 \pm 0.0048$ | $0.9166 \pm 0.0033$ | $0.6447 \pm 0.0069$ | $0.6952 \pm 0.0044$ | $0.3075 \pm 0.0090$ | $0.2381 \pm 0.0174$ |
| | TinyImageNet | $0.8743 \pm 0.0014$ | $0.8885 \pm 0.0034$ | $0.8409 \pm 0.0023$ | $0.8597 \pm 0.0027$ | $0.3862 \pm 0.0141$ | $0.3964 \pm 0.0256$ |
| | Places365 | $0.8758 \pm 0.0033$ | $0.8989 \pm 0.0037$ | $0.9497 \pm 0.0015$ | $0.9587 \pm 0.0008$ | $0.3745 \pm 0.0045$ | $0.3299 \pm 0.0246$ |
| | Average | $0.8845 \pm 0.0370$ | $0.9150 \pm 0.0197$ | $0.8194 \pm 0.1126$ | $0.8532 \pm 0.0886$ | $0.3034 \pm 0.0963$ | $0.2321 \pm 0.1030$ |
| CIFAR-100 | Gaussian Noise | $0.7191 \pm 0.1088$ | $0.7714 \pm 0.0756$ | $0.6208 \pm 0.1057$ | $0.6593 \pm 0.0951$ | $0.4667 \pm 0.1039$ | $0.3616 \pm 0.0481$ |
| | Uniform Noise | $0.7160 \pm 0.0911$ | $0.7453 \pm 0.0772$ | $0.6050 \pm 0.0981$ | $0.6239 \pm 0.0891$ | $0.4393 \pm 0.0741$ | $0.3772 \pm 0.0528$ |
| | SVHN | $0.7501 \pm 0.0355$ | $0.7659 \pm 0.0302$ | $0.8560 \pm 0.0211$ | $0.8595 \pm 0.0189$ | $0.5912 \pm 0.0570$ | $0.5370 \pm 0.0398$ |
| | Textures | $0.6879 \pm 0.0031$ | $0.7176 \pm 0.0029$ | $0.5031 \pm 0.0046$ | $0.5205 \pm 0.0048$ | $0.7566 \pm 0.0110$ | $0.6385 \pm 0.0068$ |
| | LSUN | $0.7138 \pm 0.0018$ | $0.7212 \pm 0.0018$ | $0.3746 \pm 0.0031$ | $0.3804 \pm 0.0038$ | $0.6931 \pm 0.0059$ | $0.6876 \pm 0.0041$ |
| | TinyImageNet | $0.7494 \pm 0.0032$ | $0.7660 \pm 0.0029$ | $0.7069 \pm 0.0042$ | $0.7238 \pm 0.0041$ | $0.6393 \pm 0.0037$ | $0.6166 \pm 0.0082$ |
| | Places365 | $0.7326 \pm 0.0029$ | $0.7434 \pm 0.0035$ | $0.8885 \pm 0.0018$ | $0.8938 \pm 0.0020$ | $0.6720 \pm 0.0035$ | $0.6782 \pm 0.0096$ |
| | Average | $0.7242 \pm 0.0203$ | $0.7473 \pm 0.0202$ | $0.6507 \pm 0.1702$ | $0.6659 \pm 0.1679$ | $0.6083 \pm 0.1090$ | $0.5566 \pm 0.1270$ |
| SVHN | Gaussian Noise | $0.8602 \pm 0.0706$ | $0.9279 \pm 0.0285$ | $0.6567 \pm 0.1414$ | $0.7550 \pm 0.0983$ | $0.4207 \pm 0.1935$ | $0.1750 \pm 0.0543$ |
| | Uniform Noise | $0.8330 \pm 0.0759$ | $0.8903 \pm 0.0401$ | $0.6174 \pm 0.1388$ | $0.6862 \pm 0.1053$ | $0.4974 \pm 0.1963$ | $0.3064 \pm 0.0954$ |
| | CIFAR-100 | $0.9285 \pm 0.0031$ | $0.9567 \pm 0.0015$ | $0.8072 \pm 0.0085$ | $0.8648 \pm 0.0053$ | $0.2549 \pm 0.0112$ | $0.1302 \pm 0.0051$ |
| | Textures | $0.9019 \pm 0.0142$ | $0.9319 \pm 0.0093$ | $0.6643 \pm 0.0299$ | $0.7248 \pm 0.0256$ | $0.4037 \pm 0.0827$ | $0.2299 \pm 0.0444$ |
| | LSUN | $0.9176 \pm 0.0063$ | $0.9521 \pm 0.0023$ | $0.5419 \pm 0.0200$ | $0.6445 \pm 0.0172$ | $0.2964 \pm 0.0263$ | $0.1401 \pm 0.0070$ |
| | TinyImageNet | $0.9339 \pm 0.0035$ | $0.9589 \pm 0.0016$ | $0.8160 \pm 0.0091$ | $0.8711 \pm 0.0058$ | $0.2300 \pm 0.0146$ | $0.1234 \pm 0.0051$ |
| | Places365 | $0.9266 \pm 0.0052$ | $0.9560 \pm 0.0022$ | $0.9325 \pm 0.0047$ | $0.9565 \pm 0.0025$ | $0.2607 \pm 0.0220$ | $0.1323 \pm 0.0056$ |
| | Average | $0.9003 \pm 0.0360$ | $0.9391 \pm 0.0230$ | $0.7194 \pm 0.1261$ | $0.7861 \pm 0.1050$ | $0.3377 \pm 0.0948$ | $0.1768 \pm 0.0632$ |
| TinyImageNet | Gaussian Noise | $0.7089 \pm 0.1240$ | $0.7103 \pm 0.1186$ | $0.6584 \pm 0.1297$ | $0.6510 \pm 0.1217$ | $0.6331 \pm 0.1384$ | $0.6200 \pm 0.1347$ |
| | Uniform Noise | $0.3133 \pm 0.1290$ | $0.3151 \pm 0.1270$ | $0.3841 \pm 0.0419$ | $0.3840 \pm 0.0409$ | $0.8869 \pm 0.0472$ | $0.8792 \pm 0.0479$ |
| | SVHN | $0.6870 \pm 0.0422$ | $0.6939 \pm 0.0414$ | $0.8233 \pm 0.0262$ | $0.8203 \pm 0.0269$ | $0.7291 \pm 0.0445$ | $0.7066 \pm 0.0428$ |
| | CIFAR-100 | $0.5934 \pm 0.0033$ | $0.5937 \pm 0.0031$ | $0.5644 \pm 0.0040$ | $0.5634 \pm 0.0037$ | $0.8622 \pm 0.0031$ | $0.8629 \pm 0.0035$ |
| | Textures | $0.5610 \pm 0.0109$ | $0.5651 \pm 0.0106$ | $0.3913 \pm 0.0093$ | $0.3898 \pm 0.0089$ | $0.8809 \pm 0.0058$ | $0.8645 \pm 0.0034$ |
| | LSUN | $0.5892 \pm 0.0047$ | $0.5907 \pm 0.0052$ | $0.2690 \pm 0.0044$ | $0.2671 \pm 0.0041$ | $0.8426 \pm 0.0057$ | $0.8374 \pm 0.0063$ |
| | Places365 | $0.6137 \pm 0.0029$ | $0.6152 \pm 0.0033$ | $0.8316 \pm 0.0013$ | $0.8304 \pm 0.0014$ | $0.8271 \pm 0.0025$ | $0.8210 \pm 0.0022$ |
| | Average | $0.5809 \pm 0.1201$ | $0.5834 \pm 0.1206$ | $0.5603 \pm 0.2060$ | $0.5580 \pm 0.2053$ | $0.8088 \pm 0.0869$ | $0.7988 \pm 0.0905$ |
| Average | | $0.7725 \pm 0.1461$ | $0.7962 \pm 0.1566$ | $0.6875 \pm 0.1843$ | $0.7158 \pm 0.1874$ | $0.5146 \pm 0.2285$ | $0.4411 \pm 0.2710$ |

Table 8: Expanded results for Maximum Softmax Probability detector comparing Intra-class mixup and Intra-class mixup with angular margin.

| | | ODIN | | | | | |
|---|---|---|---|---|---|---|---|
| | | AUROC | | AUPR | | FPR95 | |
| In-Dataset | OoD Dataset | Intra | Intra + Cos($\theta$) | Intra | Intra + Cos($\theta$) | Intra | Intra + Cos($\theta$) |
| CIFAR-10 | Gaussian Noise | $0.8898 \pm 0.0665$ | $0.9435 \pm 0.0385$ | $0.8038 \pm 0.1183$ | $0.8910 \pm 0.0764$ | $0.2091 \pm 0.1046$ | $0.1194 \pm 0.0705$ |
| | Uniform Noise | $0.9738 \pm 0.0105$ | $0.9842 \pm 0.0097$ | $0.9453 \pm 0.0287$ | $0.9640 \pm 0.0249$ | $0.0581 \pm 0.0173$ | $0.0367 \pm 0.0196$ |
| | SVHN | $0.9598 \pm 0.0116$ | $0.9572 \pm 0.0153$ | $0.9801 \pm 0.0057$ | $0.9785 \pm 0.0073$ | $0.1498 \pm 0.0388$ | $0.1560 \pm 0.0524$ |
| | Textures | $0.8927 \pm 0.0128$ | $0.9098 \pm 0.0102$ | $0.8078 \pm 0.0188$ | $0.8336 \pm 0.0145$ | $0.3799 \pm 0.0437$ | $0.3296 \pm 0.0407$ |
| | LSUN | $0.9243 \pm 0.0051$ | $0.9309 \pm 0.0047$ | $0.7552 \pm 0.0123$ | $0.7715 \pm 0.0121$ | $0.2745 \pm 0.0233$ | $0.2541 \pm 0.0180$ |
| | TinyImageNet | $0.9010 \pm 0.0068$ | $0.9036 \pm 0.0076$ | $0.8867 \pm 0.0074$ | $0.8898 \pm 0.0079$ | $0.3857 \pm 0.0302$ | $0.3750 \pm 0.0308$ |
| | Places365 | $0.9086 \pm 0.0004$ | $0.9139 \pm 0.0019$ | $0.9672 \pm 0.0004$ | $0.9691 \pm 0.0009$ | $0.3533 \pm 0.0036$ | $0.3329 \pm 0.0060$ |
| | Average | $0.9214 \pm 0.0308$ | $0.9347 \pm 0.0269$ | $0.8780 \pm 0.0832$ | $0.8996 \pm 0.0719$ | $0.2587 \pm 0.1162$ | $0.2291 \pm 0.1178$ |
| CIFAR-100 | Gaussian Noise | $0.9365 \pm 0.0374$ | $0.9878 \pm 0.0087$ | $0.8727 \pm 0.0757$ | $0.9738 \pm 0.0226$ | $0.1305 \pm 0.0661$ | $0.0298 \pm 0.0179$ |
| | Uniform Noise | $0.8070 \pm 0.1008$ | $0.9357 \pm 0.0335$ | $0.6811 \pm 0.1122$ | $0.8518 \pm 0.0701$ | $0.2741 \pm 0.1211$ | $0.1055 \pm 0.0479$ |
| | SVHN | $0.8147 \pm 0.0247$ | $0.8135 \pm 0.0188$ | $0.8859 \pm 0.0201$ | $0.8781 \pm 0.0175$ | $0.4685 \pm 0.0363$ | $0.4592 \pm 0.0292$ |
| | Textures | $0.7080 \pm 0.0094$ | $0.7679 \pm 0.0070$ | $0.5214 \pm 0.0095$ | $0.5908 \pm 0.0082$ | $0.7478 \pm 0.0202$ | $0.6369 \pm 0.0209$ |
| | LSUN | $0.7625 \pm 0.0024$ | $0.7478 \pm 0.0040$ | $0.4376 \pm 0.0036$ | $0.4138 \pm 0.0063$ | $0.6831 \pm 0.0113$ | $0.6883 \pm 0.0078$ |
| | TinyImageNet | $0.7790 \pm 0.0012$ | $0.7804 \pm 0.0008$ | $0.7376 \pm 0.0016$ | $0.7393 \pm 0.0006$ | $0.6370 \pm 0.0050$ | $0.6343 \pm 0.0076$ |
| | Places365 | $0.7778 \pm 0.0028$ | $0.7658 \pm 0.0043$ | $0.9104 \pm 0.0015$ | $0.9048 \pm 0.0022$ | $0.6624 \pm 0.0059$ | $0.6788 \pm 0.0091$ |
| | Average | $0.7979 \pm 0.0651$ | $0.8284 \pm 0.0874$ | $0.7210 \pm 0.1724$ | $0.7646 \pm 0.1843$ | $0.5148 \pm 0.2162$ | $0.4618 \pm 0.2598$ |
| SVHN | Gaussian Noise | $0.8902 \pm 0.0627$ | $0.9764 \pm 0.0115$ | $0.7592 \pm 0.1302$ | $0.9265 \pm 0.0353$ | $0.4052 \pm 0.2158$ | $0.0897 \pm 0.0411$ |
| | Uniform Noise | $0.5604 \pm 0.1762$ | $0.7152 \pm 0.0749$ | $0.3311 \pm 0.1563$ | $0.4173 \pm 0.0785$ | $0.7954 \pm 0.2306$ | $0.6907 \pm 0.1181$ |
| | CIFAR-100 | $0.9264 \pm 0.0098$ | $0.9815 \pm 0.0016$ | $0.8444 \pm 0.0178$ | $0.9531 \pm 0.0037$ | $0.3507 \pm 0.0491$ | $0.0784 \pm 0.0078$ |
| | Textures | $0.8539 \pm 0.0330$ | $0.9167 \pm 0.0139$ | $0.6394 \pm 0.0548$ | $0.7835 \pm 0.0252$ | $0.6461 \pm 0.1253$ | $0.4404 \pm 0.0766$ |
| | LSUN | $0.9081 \pm 0.0180$ | $0.9819 \pm 0.0015$ | $0.6110 \pm 0.0497$ | $0.8683 \pm 0.0112$ | $0.4339 \pm 0.0754$ | $0.0781 \pm 0.0091$ |
| | TinyImageNet | $0.9276 \pm 0.0117$ | $0.9810 \pm 0.0018$ | $0.8451 \pm 0.0213$ | $0.9523 \pm 0.0041$ | $0.3414 \pm 0.0593$ | $0.0798 \pm 0.0074$ |
| | Places365 | $0.9168 \pm 0.0160$ | $0.9794 \pm 0.0019$ | $0.9355 \pm 0.0121$ | $0.9831 \pm 0.0016$ | $0.3873 \pm 0.0763$ | $0.0881 \pm 0.0089$ |
| | Average | $0.8548 \pm 0.1225$ | $0.9331 \pm 0.0916$ | $0.7094 \pm 0.1882$ | $0.8406 \pm 0.1836$ | $0.4800 \pm 0.1601$ | $0.2208 \pm 0.2281$ |
| TinyImageNet | Gaussian Noise | $0.9772 \pm 0.0219$ | $0.9462 \pm 0.0364$ | $0.9737 \pm 0.0263$ | $0.9386 \pm 0.0448$ | $0.0943 \pm 0.0854$ | $0.2069 \pm 0.0996$ |
| | Uniform Noise | $0.5562 \pm 0.1663$ | $0.4426 \pm 0.1369$ | $0.5037 \pm 0.0968$ | $0.4370 \pm 0.0609$ | $0.6776 \pm 0.1424$ | $0.7706 \pm 0.0794$ |
| | SVHN | $0.8180 \pm 0.0125$ | $0.7840 \pm 0.0230$ | $0.9070 \pm 0.0054$ | $0.8864 \pm 0.0117$ | $0.5611 \pm 0.0282$ | $0.6242 \pm 0.0485$ |
| | CIFAR-100 | $0.6162 \pm 0.0032$ | $0.6059 \pm 0.0018$ | $0.5940 \pm 0.0032$ | $0.5830 \pm 0.0030$ | $0.8576 \pm 0.0048$ | $0.8621 \pm 0.0035$ |
| | Textures | $0.5700 \pm 0.0098$ | $0.5789 \pm 0.0065$ | $0.4415 \pm 0.0132$ | $0.4387 \pm 0.0094$ | $0.9140 \pm 0.0050$ | $0.8929 \pm 0.0054$ |
| | LSUN | $0.6045 \pm 0.0052$ | $0.6225 \pm 0.0076$ | $0.2803 \pm 0.0051$ | $0.2896 \pm 0.0050$ | $0.8350 \pm 0.0061$ | $0.8032 \pm 0.0062$ |
| | Places365 | $0.6216 \pm 0.0030$ | $0.6158 \pm 0.0050$ | $0.8344 \pm 0.0016$ | $0.8302 \pm 0.0025$ | $0.8232 \pm 0.0041$ | $0.8217 \pm 0.0026$ |
| | Average | $0.6805 \pm 0.1454$ | $0.6566 \pm 0.1499$ | $0.6478 \pm 0.2419$ | $0.6291 \pm 0.2370$ | $0.6804 \pm 0.2641$ | $0.7116 \pm 0.2210$ |
| Average | | $0.8137 \pm 0.1347$ | $0.8382 \pm 0.1505$ | $0.7390 \pm 0.1996$ | $0.7835 \pm 0.2061$ | $0.4835 \pm 0.2479$ | $0.4058 \pm 0.2934$ |

Table 9: Expanded results for ODIN comparing Intra-class mixup and Intra-class mixup with angular margin.

| In-Dataset | OoD Dataset | Energy Score AUROC | | AUPR | | FPR95 | |
| | | Intra | Intra + Cos($\theta$) | Intra | Intra + Cos($\theta$) | Intra | Intra + Cos($\theta$) |
|---|---|---|---|---|---|---|---|
| CIFAR-10 | Gaussian Noise | $0.8390 \pm 0.0910$ | $0.9076 \pm 0.0591$ | $0.7318 \pm 0.1090$ | $0.8164 \pm 0.0887$ | $0.2829 \pm 0.1262$ | $0.1654 \pm 0.0900$ |
| | Uniform Noise | $0.9249 \pm 0.0402$ | $0.9472 \pm 0.0299$ | $0.8322 \pm 0.0862$ | $0.8710 \pm 0.0696$ | $0.1192 \pm 0.0544$ | $0.0864 \pm 0.0416$ |
| | SVHN | $0.9266 \pm 0.0111$ | $0.9272 \pm 0.0116$ | $0.9422 \pm 0.0088$ | $0.9432 \pm 0.0096$ | $0.1839 \pm 0.0294$ | $0.1812 \pm 0.0286$ |
| | Textures | $0.8742 \pm 0.0099$ | $0.8961 \pm 0.0083$ | $0.7548 \pm 0.0152$ | $0.7807 \pm 0.0170$ | $0.4320 \pm 0.0437$ | $0.3380 \pm 0.0353$ |
| | LSUN | $0.9346 \pm 0.0052$ | $0.9356 \pm 0.0054$ | $0.7747 \pm 0.0169$ | $0.7740 \pm 0.0159$ | $0.2343 \pm 0.0136$ | $0.2273 \pm 0.0162$ |
| | TinyImageNet | $0.9057 \pm 0.0044$ | $0.9060 \pm 0.0039$ | $0.8898 \pm 0.0049$ | $0.8898 \pm 0.0043$ | $0.3584 \pm 0.0197$ | $0.3581 \pm 0.0233$ |
| | Places365 | $0.9156 \pm 0.0040$ | $0.9166 \pm 0.0043$ | $0.9693 \pm 0.0016$ | $0.9693 \pm 0.0016$ | $0.3225 \pm 0.0126$ | $0.3128 \pm 0.0176$ |
| | Average | $0.9029 \pm 0.0319$ | $0.9195 \pm 0.0168$ | $0.8421 \pm 0.0870$ | $0.8635 \pm 0.0711$ | $0.2762 \pm 0.0987$ | $0.2384 \pm 0.0938$ |
| CIFAR-100 | Gaussian Noise | $0.8128 \pm 0.0815$ | $0.8435 \pm 0.0692$ | $0.6856 \pm 0.0920$ | $0.7199 \pm 0.0879$ | $0.2883 \pm 0.0914$ | $0.2450 \pm 0.0788$ |
| | Uniform Noise | $0.6778 \pm 0.0991$ | $0.7311 \pm 0.0811$ | $0.5565 \pm 0.0740$ | $0.5951 \pm 0.0717$ | $0.4136 \pm 0.1134$ | $0.3513 \pm 0.0946$ |
| | SVHN | $0.7686 \pm 0.0230$ | $0.7724 \pm 0.0240$ | $0.8269 \pm 0.0137$ | $0.8312 \pm 0.0146$ | $0.4757 \pm 0.0466$ | $0.4729 \pm 0.0481$ |
| | Textures | $0.6860 \pm 0.0087$ | $0.7078 \pm 0.0068$ | $0.4850 \pm 0.0068$ | $0.5022 \pm 0.0061$ | $0.7586 \pm 0.0162$ | $0.7043 \pm 0.0102$ |
| | LSUN | $0.7295 \pm 0.0038$ | $0.7278 \pm 0.0040$ | $0.3775 \pm 0.0054$ | $0.3771 \pm 0.0054$ | $0.6662 \pm 0.0075$ | $0.6719 \pm 0.0072$ |
| | TinyImageNet | $0.7942 \pm 0.0010$ | $0.7942 \pm 0.0011$ | $0.7594 \pm 0.0027$ | $0.7593 \pm 0.0028$ | $0.6150 \pm 0.0039$ | $0.6171 \pm 0.0050$ |
| | Places365 | $0.7665 \pm 0.0041$ | $0.7644 \pm 0.0041$ | $0.9043 \pm 0.0025$ | $0.9037 \pm 0.0026$ | $0.6574 \pm 0.0082$ | $0.6643 \pm 0.0087$ |
| | Average | $0.7479 \pm 0.0481$ | $0.7630 \pm 0.0428$ | $0.6565 \pm 0.1769$ | $0.6698 \pm 0.1735$ | $0.5535 \pm 0.1537$ | $0.5324 \pm 0.1658$ |
| SVHN | Gaussian Noise | $0.8406 \pm 0.0894$ | $0.9078 \pm 0.0500$ | $0.6500 \pm 0.1680$ | $0.7647 \pm 0.1187$ | $0.4952 \pm 0.2378$ | $0.3170 \pm 0.1596$ |
| | Uniform Noise | $0.8251 \pm 0.0875$ | $0.8685 \pm 0.0614$ | $0.6260 \pm 0.1636$ | $0.6893 \pm 0.1276$ | $0.5352 \pm 0.2273$ | $0.4348 \pm 0.1837$ |
| | CIFAR-100 | $0.9122 \pm 0.0101$ | $0.9521 \pm 0.0039$ | $0.7858 \pm 0.0208$ | $0.8773 \pm 0.0085$ | $0.3605 \pm 0.0459$ | $0.1918 \pm 0.0143$ |
| | Textures | $0.8639 \pm 0.0245$ | $0.9126 \pm 0.0158$ | $0.6281 \pm 0.0447$ | $0.7269 \pm 0.0341$ | $0.6217 \pm 0.1184$ | $0.3978 \pm 0.0896$ |
| | LSUN | $0.8928 \pm 0.0210$ | $0.9423 \pm 0.0092$ | $0.5053 \pm 0.0525$ | $0.6789 \pm 0.0357$ | $0.4416 \pm 0.0824$ | $0.2412 \pm 0.0371$ |
| | TinyImageNet | $0.9197 \pm 0.0117$ | $0.9563 \pm 0.0048$ | $0.7994 \pm 0.0244$ | $0.8862 \pm 0.0107$ | $0.3210 \pm 0.0475$ | $0.1729 \pm 0.0183$ |
| | Places365 | $0.9111 \pm 0.0164$ | $0.9513 \pm 0.0072$ | $0.9235 \pm 0.0134$ | $0.9590 \pm 0.0058$ | $0.3668 \pm 0.0740$ | $0.1980 \pm 0.0310$ |
| | Average | $0.8808 \pm 0.0350$ | $0.9273 \pm 0.0300$ | $0.7026 \pm 0.1300$ | $0.7975 \pm 0.1015$ | $0.4489 \pm 0.1003$ | $0.2791 \pm 0.0976$ |
| TinyImageNet | Gaussian Noise | $0.8491 \pm 0.1009$ | $0.7804 \pm 0.1048$ | $0.7936 \pm 0.1324$ | $0.7042 \pm 0.1267$ | $0.3730 \pm 0.1908$ | $0.4802 \pm 0.1457$ |
| | Uniform Noise | $0.4347 \pm 0.1476$ | $0.3757 \pm 0.1311$ | $0.4313 \pm 0.0622$ | $0.4043 \pm 0.0469$ | $0.7789 \pm 0.1042$ | $0.8095 \pm 0.0831$ |
| | SVHN | $0.8127 \pm 0.0385$ | $0.7603 \pm 0.0384$ | $0.8950 \pm 0.0267$ | $0.8500 \pm 0.0286$ | $0.5301 \pm 0.0585$ | $0.5699 \pm 0.0509$ |
| | CIFAR-100 | $0.5958 \pm 0.0032$ | $0.5929 \pm 0.0029$ | $0.5691 \pm 0.0028$ | $0.5631 \pm 0.0032$ | $0.8663 \pm 0.0029$ | $0.8642 \pm 0.0041$ |
| | Textures | $0.5874 \pm 0.0114$ | $0.5757 \pm 0.0106$ | $0.4200 \pm 0.0131$ | $0.3936 \pm 0.0099$ | $0.8650 \pm 0.0120$ | $0.8494 \pm 0.0083$ |
| | LSUN | $0.5598 \pm 0.0069$ | $0.5736 \pm 0.0066$ | $0.2461 \pm 0.0048$ | $0.2525 \pm 0.0044$ | $0.8457 \pm 0.0085$ | $0.8335 \pm 0.0084$ |
| | Places365 | $0.6087 \pm 0.0037$ | $0.6105 \pm 0.0039$ | $0.8265 \pm 0.0016$ | $0.8257 \pm 0.0017$ | $0.8189 \pm 0.0045$ | $0.8113 \pm 0.0040$ |
| | Average | $0.6355 \pm 0.1350$ | $0.6099 \pm 0.1250$ | $0.5974 \pm 0.2277$ | $0.5705 \pm 0.2142$ | $0.7254 \pm 0.1804$ | $0.7454 \pm 0.1426$ |
| Average | | $0.7918 \pm 0.1318$ | $0.8049 \pm 0.1471$ | $0.6996 \pm 0.1872$ | $0.7253 \pm 0.1889$ | $0.5010 \pm 0.2135$ | $0.4488 \pm 0.2420$ |

Table 10: Expanded results for Energy Score comparing Intra-class mixup and Intra-class mixup with angular margin.

| In-Dist | AUROC | | | AUPR | | | FPR95 | | |
| | ERM | Inter | Intra | ERM | Inter | Intra | ERM | Inter | Intra |
|---|---|---|---|---|---|---|---|---|---|
| CIFAR-10 | $0.8370 \pm 0.0166$ | $0.8785 \pm 0.0031$ | $0.8478 \pm 0.0054$ | $0.8145 \pm 0.0082$ | $0.8407 \pm 0.0032$ | $0.8386 \pm 0.0066$ | $0.6734 \pm 0.1512$ | $0.3558 \pm 0.0092$ | $0.6792 \pm 0.0000$ |

Table 11: CIFAR-10 (In-Dist) vs CIFAR-100 (OoD) detection results for Maximum Softmax Probability detector.

| Maximum Softmax Probability Detector with Supplemental Angular Margin trained with ERM | | | | | | | |
|---|---|---|---|---|---|---|---|
| In-Dataset | OoD Dataset | AUROC | | AUPR | | FPR95 | |
| | | ERM | ERM + $\cos(\theta)$ | ERM | ERM + $\cos(\theta)$ | ERM | ERM + $\cos(\theta)$ |
| CIFAR-10 | Gaussian Noise | $0.6991 \pm 0.2232$ | $0.8546 \pm 0.0843$ | $0.6451 \pm 0.2250$ | $0.7479 \pm 0.1313$ | $0.6103 \pm 0.3818$ | $0.2192 \pm 0.1015$ |
| | Uniform Noise | $0.8722 \pm 0.0308$ | $0.8925 \pm 0.0268$ | $0.8005 \pm 0.0413$ | $0.8066 \pm 0.0473$ | $0.3513 \pm 0.2029$ | $0.2015 \pm 0.0560$ |
| | SVHN | $0.8111 \pm 0.0901$ | $0.8961 \pm 0.0249$ | $0.9005 \pm 0.0402$ | $0.9300 \pm 0.0185$ | $0.6102 \pm 0.2288$ | $0.2339 \pm 0.0399$ |
| | Textures | $0.8448 \pm 0.0089$ | $0.8994 \pm 0.0101$ | $0.7311 \pm 0.0058$ | $0.7754 \pm 0.0091$ | $0.6698 \pm 0.1196$ | $0.2940 \pm 0.0399$ |
| | LSUN | $0.8598 \pm 0.0177$ | $0.9050 \pm 0.0041$ | $0.6262 \pm 0.0203$ | $0.6821 \pm 0.0117$ | $0.6217 \pm 0.1507$ | $0.3098 \pm 0.0216$ |
| | TinyImageNet | $0.8344 \pm 0.0143$ | $0.8778 \pm 0.0041$ | $0.8169 \pm 0.0078$ | $0.8471 \pm 0.0044$ | $0.7070 \pm 0.1248$ | $0.4295 \pm 0.0077$ |
| | Places365 | $0.8422 \pm 0.0155$ | $0.8879 \pm 0.0041$ | $0.9418 \pm 0.0040$ | $0.9554 \pm 0.0013$ | $0.6851 \pm 0.1368$ | $0.3935 \pm 0.0210$ |
| | Average | $0.8234 \pm 0.0538$ | $0.8876 \pm 0.0157$ | $0.7803 \pm 0.1113$ | $0.8206 \pm 0.0907$ | $0.6079 \pm 0.1106$ | $0.2974 \pm 0.0812$ |
| CIFAR-100 | Gaussian Noise | $0.2596 \pm 0.0665$ | $0.3761 \pm 0.0693$ | $0.3647 \pm 0.0181$ | $0.4026 \pm 0.0242$ | $0.8237 \pm 0.0798$ | $0.6752 \pm 0.0729$ |
| | Uniform Noise | $0.7941 \pm 0.0627$ | $0.8030 \pm 0.0547$ | $0.6889 \pm 0.0805$ | $0.6894 \pm 0.0709$ | $0.3759 \pm 0.0618$ | $0.3441 \pm 0.0441$ |
| | SVHN | $0.7426 \pm 0.0273$ | $0.7686 \pm 0.0187$ | $0.8507 \pm 0.0173$ | $0.8597 \pm 0.0144$ | $0.6120 \pm 0.0456$ | $0.5155 \pm 0.0118$ |
| | Textures | $0.6948 \pm 0.0038$ | $0.7200 \pm 0.0046$ | $0.5075 \pm 0.0046$ | $0.5242 \pm 0.0048$ | $0.7322 \pm 0.0079$ | $0.6472 \pm 0.0105$ |
| | LSUN | $0.7020 \pm 0.0051$ | $0.7101 \pm 0.0048$ | $0.3591 \pm 0.0061$ | $0.3666 \pm 0.0058$ | $0.7094 \pm 0.0059$ | $0.7182 \pm 0.0063$ |
| | TinyImageNet | $0.7382 \pm 0.0026$ | $0.7577 \pm 0.0023$ | $0.6900 \pm 0.0042$ | $0.7092 \pm 0.0038$ | $0.6518 \pm 0.0025$ | $0.6239 \pm 0.0047$ |
| | Places365 | $0.7201 \pm 0.0040$ | $0.7327 \pm 0.0035$ | $0.8818 \pm 0.0022$ | $0.8881 \pm 0.0020$ | $0.6992 \pm 0.0066$ | $0.7068 \pm 0.0036$ |
| | Average | $0.6645 \pm 0.1680$ | $0.6955 \pm 0.1336$ | $0.6204 \pm 0.1990$ | $0.6343 \pm 0.1932$ | $0.6577 \pm 0.1304$ | $0.6044 \pm 0.1233$ |
| SVHN | Gaussian Noise | $0.9218 \pm 0.0141$ | $0.9376 \pm 0.0102$ | $0.7770 \pm 0.0298$ | $0.8058 \pm 0.0263$ | $0.2575 \pm 0.0560$ | $0.1884 \pm 0.0338$ |
| | Uniform Noise | $0.9247 \pm 0.0061$ | $0.9391 \pm 0.0038$ | $0.7845 \pm 0.0061$ | $0.8097 \pm 0.0065$ | $0.2494 \pm 0.0335$ | $0.1863 \pm 0.0186$ |
| | CIFAR-100 | $0.9267 \pm 0.0042$ | $0.9416 \pm 0.0023$ | $0.7993 \pm 0.0067$ | $0.8303 \pm 0.0050$ | $0.2552 \pm 0.0257$ | $0.1919 \pm 0.0114$ |
| | Textures | $0.9066 \pm 0.0059$ | $0.9250 \pm 0.0053$ | $0.6651 \pm 0.0154$ | $0.7068 \pm 0.0150$ | $0.3723 \pm 0.0364$ | $0.2685 \pm 0.0310$ |
| | LSUN | $0.9258 \pm 0.0051$ | $0.9409 \pm 0.0033$ | $0.5614 \pm 0.0133$ | $0.6108 \pm 0.0136$ | $0.2534 \pm 0.0260$ | $0.1912 \pm 0.0155$ |
| | TinyImageNet | $0.9308 \pm 0.0037$ | $0.9453 \pm 0.0022$ | $0.8071 \pm 0.0066$ | $0.8392 \pm 0.0050$ | $0.2306 \pm 0.0221$ | $0.1738 \pm 0.0121$ |
| | Places365 | $0.9281 \pm 0.0047$ | $0.9442 \pm 0.0030$ | $0.9324 \pm 0.0034$ | $0.9460 \pm 0.0026$ | $0.2436 \pm 0.0256$ | $0.1781 \pm 0.0138$ |
| | Average | $0.9235 \pm 0.0074$ | $0.9391 \pm 0.0063$ | $0.7610 \pm 0.1088$ | $0.7927 \pm 0.0986$ | $0.2660 \pm 0.0442$ | $0.1969 \pm 0.0299$ |
| TinyImageNet | Gaussian Noise | $0.5506 \pm 0.1104$ | $0.5598 \pm 0.1041$ | $0.4911 \pm 0.0665$ | $0.4941 \pm 0.0638$ | $0.7374 \pm 0.1040$ | $0.7174 \pm 0.1011$ |
| | Uniform Noise | $0.2977 \pm 0.0686$ | $0.3028 \pm 0.0708$ | $0.3726 \pm 0.0220$ | $0.3743 \pm 0.0230$ | $0.9064 \pm 0.0255$ | $0.8966 \pm 0.0242$ |
| | SVHN | $0.6747 \pm 0.0258$ | $0.6806 \pm 0.0269$ | $0.7956 \pm 0.0237$ | $0.7955 \pm 0.0240$ | $0.7187 \pm 0.0201$ | $0.7025 \pm 0.0201$ |
| | CIFAR-100 | $0.5774 \pm 0.0034$ | $0.5775 \pm 0.0035$ | $0.5491 \pm 0.0038$ | $0.5486 \pm 0.0040$ | $0.8718 \pm 0.0035$ | $0.8730 \pm 0.0032$ |
| | Textures | $0.5555 \pm 0.0078$ | $0.5609 \pm 0.0078$ | $0.3801 \pm 0.0099$ | $0.3818 \pm 0.0092$ | $0.8702 \pm 0.0056$ | $0.8580 \pm 0.0046$ |
| | LSUN | $0.5897 \pm 0.0051$ | $0.5901 \pm 0.0048$ | $0.2709 \pm 0.0041$ | $0.2697 \pm 0.0042$ | $0.8451 \pm 0.0044$ | $0.8437 \pm 0.0051$ |
| | Places365 | $0.6117 \pm 0.0045$ | $0.6134 \pm 0.0037$ | $0.8309 \pm 0.0014$ | $0.8308 \pm 0.0015$ | $0.8334 \pm 0.0042$ | $0.8288 \pm 0.0041$ |
| | Average | $0.5510 \pm 0.1104$ | $0.5550 \pm 0.1099$ | $0.5272 \pm 0.1990$ | $0.5278 \pm 0.1987$ | $0.8261 \pm 0.0658$ | $0.8171 \pm 0.0708$ |
| Average | | $0.7406 \pm 0.1771$ | $0.7693 \pm 0.1764$ | $0.6722 \pm 0.1915$ | $0.6939 \pm 0.1948$ | $0.5895 \pm 0.2243$ | $0.4789 \pm 0.2600$ |

Table 12: Expanded results for Maximum Softmax Probability Detector with Supplemental Angular Margin trained with ERM.

| Maximum Softmax Probability Detector with Supplemental Angular Margin trained with Inter-class Mixup | | | | | | | |
|---|---|---|---|---|---|---|---|
| In-Dataset | OoD Dataset | AUROC | | AUPR | | FPR95 | |
| | | Inter | Inter + $\cos(\theta)$ | Inter | Inter + $\cos(\theta)$ | Inter | Inter + $\cos(\theta)$ |
| CIFAR-10 | Gaussian Noise | $0.9470 \pm 0.0471$ | $0.9573 \pm 0.0448$ | $0.9073 \pm 0.0816$ | $0.9260 \pm 0.0777$ | $0.1290 \pm 0.1128$ | $0.1077 \pm 0.1152$ |
| | Uniform Noise | $0.9462 \pm 0.0383$ | $0.9477 \pm 0.0433$ | $0.9059 \pm 0.0621$ | $0.9116 \pm 0.0633$ | $0.1383 \pm 0.0960$ | $0.1492 \pm 0.1395$ |
| | SVHN | $0.8539 \pm 0.0386$ | $0.8523 \pm 0.0405$ | $0.9263 \pm 0.0167$ | $0.9245 \pm 0.0179$ | $0.6285 \pm 0.1962$ | $0.6131 \pm 0.1681$ |
| | Textures | $0.8471 \pm 0.0144$ | $0.8472 \pm 0.0113$ | $0.7677 \pm 0.0147$ | $0.7782 \pm 0.0117$ | $0.7438 \pm 0.0782$ | $0.7583 \pm 0.0652$ |
| | LSUN | $0.8888 \pm 0.0063$ | $0.9064 \pm 0.0061$ | $0.7074 \pm 0.0111$ | $0.7446 \pm 0.0122$ | $0.5010 \pm 0.0439$ | $0.4107 \pm 0.0451$ |
| | TinyImageNet | $0.8484 \pm 0.0046$ | $0.8441 \pm 0.0021$ | $0.8456 \pm 0.0052$ | $0.8496 \pm 0.0038$ | $0.6929 \pm 0.0187$ | $0.7481 \pm 0.0022$ |
| | Places365 | $0.8546 \pm 0.0055$ | $0.8713 \pm 0.0060$ | $0.9510 \pm 0.0018$ | $0.9570 \pm 0.0019$ | $0.6920 \pm 0.0307$ | $0.6163 \pm 0.0425$ |
| | Average | $0.8837 \pm 0.0419$ | $0.8895 \pm 0.0445$ | $0.8587 \pm 0.0837$ | $0.8702 \pm 0.0756$ | $0.5037 \pm 0.2445$ | $0.4862 \pm 0.2502$ |
| CIFAR-100 | Gaussian Noise | $0.8916 \pm 0.0579$ | $0.9012 \pm 0.0502$ | $0.7963 \pm 0.0875$ | $0.8070 \pm 0.0810$ | $0.2005 \pm 0.0778$ | $0.1832 \pm 0.0712$ |
| | Uniform Noise | $0.7464 \pm 0.2002$ | $0.7559 \pm 0.1863$ | $0.6705 \pm 0.1818$ | $0.6708 \pm 0.1714$ | $0.3751 \pm 0.2218$ | $0.3527 \pm 0.2003$ |
| | SVHN | $0.7460 \pm 0.0302$ | $0.7569 \pm 0.0315$ | $0.8423 \pm 0.0150$ | $0.8488 \pm 0.0159$ | $0.6137 \pm 0.0796$ | $0.5830 \pm 0.0761$ |
| | Textures | $0.7454 \pm 0.0030$ | $0.7571 \pm 0.0030$ | $0.5502 \pm 0.0047$ | $0.5622 \pm 0.0039$ | $0.6451 \pm 0.0133$ | $0.6032 \pm 0.0118$ |
| | LSUN | $0.7417 \pm 0.0037$ | $0.7479 \pm 0.0034$ | $0.4014 \pm 0.0076$ | $0.4065 \pm 0.0078$ | $0.7162 \pm 0.0130$ | $0.6871 \pm 0.0113$ |
| | TinyImageNet | $0.7857 \pm 0.0023$ | $0.7958 \pm 0.0022$ | $0.7447 \pm 0.0022$ | $0.7573 \pm 0.0023$ | $0.6305 \pm 0.0133$ | $0.6047 \pm 0.0109$ |
| | Places365 | $0.7560 \pm 0.0019$ | $0.7645 \pm 0.0016$ | $0.8989 \pm 0.0011$ | $0.9026 \pm 0.0011$ | $0.7035 \pm 0.0043$ | $0.6716 \pm 0.0037$ |
| | Average | $0.7733 \pm 0.0503$ | $0.7828 \pm 0.0504$ | $0.7006 \pm 0.1619$ | $0.7079 \pm 0.1619$ | $0.5549 \pm 0.1786$ | $0.5265 \pm 0.1732$ |
| SVHN | Gaussian Noise | $0.8198 \pm 0.1368$ | $0.8921 \pm 0.0841$ | $0.6935 \pm 0.2014$ | $0.7639 \pm 0.1620$ | $0.6185 \pm 0.4020$ | $0.4099 \pm 0.2984$ |
| | Uniform Noise | $0.7706 \pm 0.1228$ | $0.8136 \pm 0.1033$ | $0.6385 \pm 0.1586$ | $0.6726 \pm 0.1441$ | $0.7598 \pm 0.3313$ | $0.6637 \pm 0.2931$ |
| | CIFAR-100 | $0.9340 \pm 0.0071$ | $0.9481 \pm 0.0044$ | $0.8672 \pm 0.0108$ | $0.8875 \pm 0.0075$ | $0.3282 \pm 0.0841$ | $0.2184 \pm 0.0385$ |
| | Textures | $0.8851 \pm 0.0253$ | $0.9136 \pm 0.0181$ | $0.7391 \pm 0.0417$ | $0.7728 \pm 0.0336$ | $0.7703 \pm 0.1736$ | $0.5329 \pm 0.1445$ |
| | LSUN | $0.9353 \pm 0.0110$ | $0.9529 \pm 0.0066$ | $0.7183 \pm 0.0313$ | $0.7612 \pm 0.0241$ | $0.3332 \pm 0.1233$ | $0.1886 \pm 0.0421$ |
| | TinyImageNet | $0.9354 \pm 0.0105$ | $0.9509 \pm 0.0062$ | $0.8732 \pm 0.0176$ | $0.8952 \pm 0.0123$ | $0.3254 \pm 0.1094$ | $0.2036 \pm 0.0478$ |
| | Places365 | $0.9355 \pm 0.0133$ | $0.9521 \pm 0.0083$ | $0.9540 \pm 0.0087$ | $0.9642 \pm 0.0059$ | $0.3307 \pm 0.1388$ | $0.1968 \pm 0.0607$ |
| | Average | $0.8880 \pm 0.0624$ | $0.9176 \pm 0.0477$ | $0.7834 \pm 0.1065$ | $0.8168 \pm 0.0937$ | $0.4952 \pm 0.1967$ | $0.3448 \pm 0.1787$ |
| TinyImageNet | Gaussian Noise | $0.4041 \pm 0.1878$ | $0.4063 \pm 0.1324$ | $0.4355 \pm 0.1123$ | $0.4176 \pm 0.0562$ | $0.7842 \pm 0.1274$ | $0.7800 \pm 0.0989$ |
| | Uniform Noise | $0.3944 \pm 0.1244$ | $0.3678 \pm 0.1272$ | $0.4104 \pm 0.0488$ | $0.4008 \pm 0.0469$ | $0.7805 \pm 0.1021$ | $0.7911 \pm 0.1043$ |
| | SVHN | $0.6042 \pm 0.0915$ | $0.5479 \pm 0.0750$ | $0.7473 \pm 0.0607$ | $0.7004 \pm 0.0417$ | $0.7458 \pm 0.0843$ | $0.7622 \pm 0.0695$ |
| | CIFAR-100 | $0.5680 \pm 0.0098$ | $0.5671 \pm 0.0094$ | $0.5423 \pm 0.0095$ | $0.5393 \pm 0.0096$ | $0.8819 \pm 0.0034$ | $0.8818 \pm 0.0023$ |
| | Textures | $0.5435 \pm 0.0143$ | $0.5195 \pm 0.0093$ | $0.3756 \pm 0.0091$ | $0.3506 \pm 0.0062$ | $0.8786 \pm 0.0094$ | $0.8793 \pm 0.0089$ |
| | LSUN | $0.5727 \pm 0.0263$ | $0.5750 \pm 0.0195$ | $0.2615 \pm 0.0186$ | $0.2598 \pm 0.0112$ | $0.8639 \pm 0.0185$ | $0.8581 \pm 0.0188$ |
| | Places365 | $0.6032 \pm 0.0106$ | $0.6040 \pm 0.0148$ | $0.8270 \pm 0.0061$ | $0.8258 \pm 0.0088$ | $0.8424 \pm 0.0070$ | $0.8377 \pm 0.0085$ |
| | Average | $0.5272 \pm 0.0832$ | $0.5125 \pm 0.0835$ | $0.5142 \pm 0.1901$ | $0.4992 \pm 0.1869$ | $0.8253 \pm 0.0505$ | $0.8271 \pm 0.0455$ |
| Average | | $0.7680 \pm 0.1588$ | $0.7693 \pm 0.1764$ | $0.7142 \pm 0.1914$ | $0.7235 \pm 0.1977$ | $0.5895 \pm 0.2243$ | $0.5462 \pm 0.2500$ |

Table 13: Expanded results for Maximum Softmax Probability Detector with Supplemental Angular Margin trained with Inter-class Mixup.

