# OpenReview forum: "Intra-class Mixup for Out-of-Distribution Detection"
_ICLR.cc/2022/Conference — ICLR 2022 Submitted_

### Official Review · Reviewer_DAih · 2021-10-27

**Correctness:** 3
**Technical Novelty And Significance:** 2
**Empirical Novelty And Significance:** 2
**Recommendation:** 3
**Confidence:** 4

**Main Review:**

A strong point of the paper is that they evaluate their method in many different settings, i.e. on 3 OOD methods on 4 different in-distribution. They also report AUROC, AUPR and FPR95, which is good. They do not fit special hyperparameters for each specific out-distribution, making their comparison fair in my opinion. Furthermore they test their OOD detection performance on many different out-distributions.

That being said, one OOD detection task that they should definitely add is CIFAR10 vs. CIFAR100 and vice-versa, as this task is known to be particularly difficult.

Overall, the paper's use of the angular margin is not well-motivated. Especially the creation of a new OOD score by simply adding the angular margin to other OOD scores is completely ad-hoc. On top of this, it should be noted that their angular margin score depends on arbitrary bias values in the last layer. If one adds some bias shift $b$ to every bias in the last layer then it shifts the origin and thus modifies the angular margin. However, the resulting network's loss would be identical to the original and thus there is a-priori no reason that training should favor one network over the other. All their OOD performance measures would also remain invariant except for the angular margin score. There should at least be a discussion about why despite all this, the angular margin score is still helpful.

I am not sure about the strength of their baselines. Comparing for example with the results from [1], they used the same architecture but their simple MSP model beats even the best results in Table 9 (on all but one of the datasets and metrics that both report). This doesn't even mention the fact that even if this wasn't the case the paper would still only be comparing their results to weak baselines. Basically, the authors argue that one can improve all three of the baselines that they study using their technique. But even the improved version's performance is worse than, for example outlier exposure [Hendrycks et. al. 2019] or Mahalanobis distance [Lee et. al 2018], so why use the weak techniques in the first place?

One point of confusion for me was that the standard deviations in Table 1 were seemingly computed across the dataset, whereas in all other tables they were computed across different random seeds. I really think this could be clarified explicitly in the Table's caption (especially given that the authors had lots of space left).

I was also confused how exactly the standard deviations of the averages were computed in Table 4, Table 5 and in the appendix. Are all runs aggregated and then the standard deviation across all of them is computed? Or is the average computed for each seed and then the standard deviation of that is reported?

More issues:
- In Table 1 the caption reads "Intra-class mixup has consistently has better separability" (small typo here!), even though it clearly isn't true for CIFAR-10.
- Table 2 is fairly useless to the paper and appears to have been added to pad out the length.
- In Table 3, we see that Inter generally performs better in terms of accuracy (the gap being quite large on CIFAR-100). This should at least be acknowledged and discussed by the authors. Arguably, this issue is strong enough as an argument against using their method just by itself, so it cannot be glossed over.
- In Table 4, there are 6 instances of wrong boldfacing, making the authors' results look better than they are.

Minor comments:
- The citation style is very hard to read. Presumably the authors used \citet (instead of \citep) even for citations that are not directly referenced in the sentence.
- "Angular" is misspelled in all axis labels in Figure 1.
- All Table captions should be above the tables, not below.
- All equations are missing punctuation.
- Wenn formatting $\cos$ or $\arccos$ in mathmode one should use \cos and \arccos (instead of $cos$ and $arccos$). Similarly, I would recommend to use \mathrm for worded subscripts, i.e. $R_\mathrm{expected}$ instead of $R_{expected}$.
- Several figure and table captions end without punctuation.

[1] Towards neural networks that provably know when they don't know, Alexander Meinke, Matthias Hein, at ICLR20

**Summary Of The Paper:**

The authors propose to use intra-class mixup (i.e. linearly interpolating images from the same class) as a form of data augmentation. They further define the angular margin and propose to add it to out-of-distribution detection scores in order to improve the performance. They evaluate their method's OOD performance on three different OOD scores and show improvements.

**Summary Of The Review:**

I recommend to reject this paper. The theoretical contribution is marginal and mostly unmotivated and the empirical results only compare to relatively weak baselines. On top of this, the paper clearly lacks polish.

---

> ### Author Response · Authors · 2021-11-23
> **Authors' response**
>
> We thank the reviewer for their valuable feedback.
>
> * We exclude the bias term when calculating the angular margin. We have revised the text to say so. The reason behind the choice of adding angular margin lies in Figure 1. Which shows that using the l2 margin only considers information in one component (i.e. in the direction of the ‘n’) and completely ignores the information in the angular margin. Similarly using angular margin alone ignores the information from the l2 margin, therefore both are to be used together. We have added text (Section 4.3) in the revised version to make this clear. Further, we have added the result of using only cos(theta) to Table 3.
>
>
> * More Details:
> From Figure 1, we can interpret that higher values along the y-axis are more likely to be OoD (i.e. using only l2). Further, we see that using angular margin improves separability (using two axes instead of one). When existing techniques are provided with angular information, they perform better.  This is clearly seen in Table 4 where adding the angular information to existing techniques improves their performance.
>
>
> * We have added the result for CIFAR-10 vs CIFAR-100 in the Appendix (Table 11). And this seems to favor inter-class mixup.
>
>
> * As stated in the paper our method is unsupervised (it does not use any auxiliary dataset).  With regards to [1], outlier exposure [Hendrycks et. al. 2019] or Mahalanobis distance [Lee et. al 2018], we believe are supervised methods. Comparing our work directly with [1], outlier exposure [Hendrycks et. al. 2019] or Mahalanobis distance [Lee et. al 2018] is therefore unfair since they belong to two separate categories.
>
>
> * Regarding the results of [1] and Table 9, our evaluation uses more OoD datasets than [1] which are harder to separate such as Places365, LSUN, TinyImageNet, Textures and more. The numbers in [1] are on single seeds, while ours is averaged over 5 different initializations. Also, another factor that affects AUROC numbers is baseline accuracy of the trained network. This can be affected by different factors such as pre-processing steps used. Since no baseline accuracies are available in [1] there is missing data for direct comparison. Therefore, using Table 3, Table 4 and Figure 2 (Table numbers in the revised version) are fair comparisons. Further compared to [1] their in-distribution datasets are different. Therefore, we believe our experimental results are through and comparing the most appropriate methods.
>
>
> * More Details:
> Supervised methods assume a prior on the OoD data, while unsupervised methods do not. The prior is often in the form of auxiliary dataset. Mahalanobis distance [Lee et. al 2018] in their code learn on each test OoD dataset making their method supervised. Comparing our work directly with [1], outlier exposure [Hendrycks et. al. 2019] or Mahalanobis distance [Lee et. al 2018] is therefore unfair since they belong to two separate categories. In the unsupervised category the works compared in this paper represent few of the best performing methods.
>
> * The results in Table 1 are averaged over seeds and OoD datasets. We have revised the caption to be clearer.
>
> * For all results in the main paper and appendix the expirements were run for various seeds and on different OoD datasets. The results were aggregated and the mean and standard deviation across all of them was computed. For example, in Table 5 (revised paper table number), the “Average” below Places365 is aggregate runs over the datasets and seeds, so 7 (OoD datasets) x 5 (seeds) = 35 datapoints. The overall average for a method is the overall aggregated average and deviation of 140 (i.e. 5 (seeds) x 7 (OoD datasets) x 4 (In-dist. datasets)) datapoints. We have revised the text to be clearer.
>
> * The observation that for CIFAR-100 intra-class mixup trails inter-class mix-up is valid, but intra-class mixup is close. We have revised the caption to acknowledge this.
>
> * Table 2 has been omitted.
>
> * Improved performance of inter-class mixup on baseline has been discussed in detail in the original mixup paper, we wanted to focus our work on the OoD performance.
>
> * More Details:
> Please note that intra-class mixup has better accuracy than ERM, while it is slightly worse than inter-class mixup. The improved performance on inter-class mixup has been attributed to the soft labels and the regularization effect it has. Intra-class mixup can be viewed as a hybrid between inter-class mixup and ERM. While intra-class mixup has benefits for OoD detection, intra-class mixup reduces the effect of soft labels thus reducing the regularization effect leading to the lower performance on baseline compared to inter-class mixup.
>
> * Table 3 (intermediate results table) had wrong bold facing issues which were not intentional and have been addressed. Table 4 (final result) does not have this issue. Also, we would like to gently remind that for FPR95 lower values are better.
>
> * We have addressed the citation style and minor issues.

---

> > ### Comment · Reviewer_DAih · 2021-11-24
> > **Acknowledgement of rebuttal**
> >
> > I acknowledge the authors' response and appreciate their changes. It would have been very helpful to mark the revisions in the paper using a different color.
> >
> > I don't quite understand why Table 11 wasn't integrated directly into the main tables, besides that it might reduce the average performance of the proposed method. Also, as I stated in my original review, CIFAR-100 vs. CIFAR-10 should also be added.
> >
> > I still don't see why the simple sum of the $\cos(\theta)$-term to any other OOD score is motivated. Obviously any two weak features can be linearly combined to create a potentially stronger feature. The question is: why is this the way to combine them? Even if we took for granted that the different scores should be naively added (despite fulfilling different invariances with respect to transformations of the weights), one introduces the hyperparameter of their relative weighting. This is chosen to be 1 in this paper with no justification or ablation, as if it were completely obvious. Why is that?
> >
> > I certainly agree that outlier exposure uses additional data. However, there is an unsupervised version of [Lee et. al 2018] that calibrates on Adversarial Examples.
> >
> > Regarding the results of [1]: note that their Table 1 does, in fact, list the accuracy of all of their baselines and that they are quite close to the ones in this manuscript so this particular argument in the rebuttal in invalid. I do acknowledge and agree with the fact that the present paper analyzes the performance across different seeds and that, in principle, it is possible that the higher performance of Plain in [1] is simply a statistical fluke. It is not true, however that they can't be compared at all: some in-out-distrtibution pairs are identical.
> >
> > Regarding the wrong boldfacing: Please note that, at the time of my review, Table 3 was still called Table 4 and that I am very aware that for FPR95 lower is better. Unfortunately, wrong boldfacing still persists: MSP AUROC CF-10 Inter=Intra, Vanilla $\cos(\theta)$ AUROC CF-100 ERM is best.
> >
> > I do not believe that my criticisms have been properly addressed - most importantly the motivation of the simple sum of scores and the proper incorporation of CIFAR-10 vs. CIFAR-100. Thus, I am not raising my score at this time.

---

> > > ### Author Response · Authors · 2021-11-25
> > > **Authors' Response**
> > >
> > > We thank the reviewer for their feedback.
> > > * We have added CIFAR-10 vs. CIFAR-100 results in Table 11. Integrating Table 11 in Table 3 would require running all experiments for Table 3 (140 x 3 experiments) to get mean and standard deviation. This would not be possible within the given time constraints. Therefore, the best option for us was to provide the results that we could obtain in the given time. We will include all results in the final manuscript.
> > > * The reason for adding the scores is because we treat the two scores as two detectors and ensemble the scores together. Averaging scores (addition effectively achieves this) is the simplest approach.  More complex approaches such as weighted average could be used. Weights for weighted average would need to be obtained using an auxiliary OoD dataset. This would result in a supervised version of the proposed approach. We focus on showing that using intra-class mixup and the angular margin is worthwhile for OoD detection.
> > > * Regarding the results of [1]: Factors such as choice of preprocessing used and the size of the OoD set compared to in-dist. test set affect the AUROC, AUPR and FPR95 results. We observe (see performance on TinyImageNet) that the baseline accuracy influences AUROC. Further, please note that when using AUPR numbers we observe similar results (please refer to ODIN results) on both papers.
> > > * Regarding the [Lee et al. 2018] work, the use of adversarial examples could be also classified as supervised where the auxiliary dataset is the set of adversarial examples. Hence, we did not compare our work with [Lee et al. 2018].
> > > * Regarding the wrong boldfacing, this was unintentional, and will be updated in the final manuscript.

---

### Official Review · Reviewer_WTWp · 2021-10-31

**Correctness:** 3
**Technical Novelty And Significance:** 3
**Empirical Novelty And Significance:** 2
**Recommendation:** 5
**Confidence:** 5

**Main Review:**

Strengths
- Overall the paper is clearly written, although the experiments and results could be improved.
- Good coverage of the related work
- Very relevant topic

Weaknesses
- This work is presented as an alternative to standard empirical risk minimization (ERM) or the setup of mixup (Zhang et al, ICLR 2018), where ERM is replaced by vicinal risk minimization (VRM, Chappelle et al, NIPS 2000). What learning strategy is used here, in opposition to these two works, is never stated.
- The experimental setup is not clear, which limits reproducibility and makes difficult to interpret the obtained results. What exactly is considered as in-distribution and what samples are OoD and how are they generated? (see detailed comments)
- The proposed comparisons against ERM and mixup are not necessarily relevant. ERM is the standard approach to train a model and mixup proposes a data augmentation strategy to make a model more robust to adversarial samples. This work, instead focuses on the detection of OOD samples. As such, it should compare itself with methods addressing the same problem and not directly against ERM and mixup (table 1).

Detailed comments
- Please acknowledge previous work on angle-based outlier detection [1], as it closely relates to this work.
- The angular margin is estimated w.r.t the decision boundary (see eqs. 3 and 4). Therefore, there is an error in the illustration in Fig. 1a.
- Differently from mixup, in this work \lambda does not follow a Beta distribution. Moreover, no details are provided on how it is chosen. Please comment.
- If the label should not change, second line of Eq. 2 could be omitted. Otherwise, there is no guarantee that \hat{y} will have the same value. This would only hold when \lambda=1 or 0 and y_i=y_j=1 or 0.
- Eq 5 implies that new data is being generated, i.e. data augmentation. Is this the case? What happends with the original samples? I suppose they are undesired since they have high variance.
- Could the authors motivate why the angular margin needs to be used coupled with another metric (eq. 7) and not on its own?
- The experimental setup is not clear. The paper misses to clearly establish what is an OoD sample/set on each of the experiments. In which way Gaussian and uniform noise are used for this purpose?
- At inference time, when is a sample considered OOD?
- The AUROC is not a good measure in OOD detection problems, since usually the majority class dominates. AUPRC should be favored. Interestingly, it is mixup that fairs best in that scenario, despite not being a method designed for OOD detection. Please comment

Minor
- There are typos in the plots in figure 1b,c,d
- Table 2 is unnecessary and may be omitted

References
[1] H.-P. Kriegel, M. S hubert, and A. Zimek. Angle-based outlier detection in high-dimensional data. In Proceedings of the 14th ACM SIGKDD International Conference on Knowledge Discovery and Data Mining, KDD ’08, pages 444–452. ACM, 2008.


**Summary Of The Paper:**

Inspired from inter-class mixup (Zhang et al, ICLR 2018), where data augmentation is used to train models more robust to adversarial samples, this work proposes intra-class mixup to reduce the variance of in-distribution samples, i.e. the training set, and improve the capacity of a trained model to detect out-of-distribution samples at inference time. The key difference between the two works is that the here proposed does the mixup within samples belonging to the same class.

In addition to this, the work propose to use the angular margin, i.e. the angle between the normal of the decision boundary of a neural net (obtained from the weights of the last layer) and an unmixed sample, to detect OOD samples. The cosine of such angle shall be coupled with an OOD method to perform OOD detection.

**Summary Of The Review:**

This work proposes intra-class mixup coupled with an angular mesaure for OOD detection. The idea of an angle-based measure has been explored in the past (not mentioned in the work) and it is interesting to propose it in the context of deep nets.

The work, however, misses some key elements in the presentation of the method and has a weak experimental setup.

---

> ### Author Response · Authors · 2021-11-23
> **Authors' response**
>
> We thank the reviewer for their valuable feedback.
> * The learning strategy used in this paper is intra-class mixup and can be categorized under vicinal risk minimization (VRM). Intra-class mixup is mathematically stated in Section 4.2 under Methodology (Section 4).
> * Regarding methods addressing the same problem, Table 1 is the motivating experiment showing the properties of intra-class mixup. We provide results that compare our work with methods addressing the same problem in the subsequent sections namely in Table 4 and Figure 2, further in Tables 8, 9 and 10.
> * We have acknowledged [1] in the new revision.
>
> * We exclude the bias term when calculating angular margin, this centers the vectors at the origin. We have revised the text to say so and further we have updated the figure for ease of understanding.
>
> * In the revised version we have detailed the way beta is chosen which states, “In the same fashion as inter-class mixup lambda ~ Beta(alpha, alpha) and for our experiments we set alpha = 1”.
>
> * Since we enforce y_j = y_i for intra-class mixup any choice of lambda in [0, 1] will guarantee y_hat = y_i = y_j. Since interpolating between label values results in the same label value. For example, if lambda is 0.2 and y_i = y_j = 3, then y_hat = 3 * 0.2 + 3 * 0.8 = 3 * 1 = 3.
>
> * Similar to inter-class mixup, intra class mixup augments data but within the same class. The original samples are not used to train the network exactly as in inter-class mixup. However please note, during inference no mixup is performed same as inter-class mixup.
>
> * The motivation for coupling angular margin lies in Figure 1 which shows that using the l2 margin only considers information in one component (i.e. in the direction of the ‘n’) and completely ignores the information in the angular margin. Similarly using angular margin alone ignores the information from the l2 margin. Therefore, both are required to get the most out of the detector. We have added the result of using only cos(theta) to Table 3 (please refer to Table 3 and 4 for comparisons). We have added text (Section 4.3) in the revised version to make this clear.
>
> * More Details:
> From Figure 1, we can interpret that higher values along the y-axis are more likely to be OoD (i.e. using only l2). Further, we see that using angular margin improves separability (using two axes instead of one). When existing techniques are provided with angular information, they perform better.  This is clearly seen in Table 4 where adding the angular information to existing techniques improves their performance.
>
> * We agree with the reviewer that the experimental setup lacked clarity we have revised the paper (Section 5.2) to clarify the setup.  The literature in this domain uses data from data distributions other than the training set as OoD data. We follow the same approach. For example, if a network is trained on CIFAR-10, datasets other than CIFAR-10 are considered OoD. Also, please note that we do not use Gaussian or Uniform Noise other than for testing as OoD datasets. When using MSP detector during inference, a sample is considered OoD when the softmax confidence of the predicted class is less than a threshold.
>
> * More Details:
> In our work when we trained on CIFAR-10, datasets other than CIFAR-10 (i.e. SVHN, Places365, Textures, LSUN, TinyImageNet, Gaussian Noise and Uniform Noise) were used as OoD data. Table 3 is a condensed version of the Table 5, 6, and 7 (tables numbers in the latest revised version) which details the in-distribution and OoD datasets used. The detection performance was obtained by aggregating results from 5 differently seeded models and OoD datasets.
>
> * During inference, when is a sample considered OoD? This depends on the detection scheme. For example, when using energy-based method a sample is considered OoD when the energy score is less than a threshold. When using MSP and ODIN techniques, a sample is considered OoD when the softmax confidence of the predicted class is less than a threshold. In general, it’s a threshold on some metric during inference.
>
> *  It is true that AUROC suffers from this issue. We balance in-dist. and OoD datasets as much as possible while using all the samples. On Gaussian and Uniform Noise, we generate test sizes that match the in-distribution test set size. On average (over various methods, seeds and OoD datasets) intra-class mixup has a marginal improvement (0.2%) over inter-class mixup on AUPR scores. Further, we also provide FPR95 metrics which also shows significant improvement in the case of intra-class mixup. Thus, intra class-mixup has better avg. performance on all the 3 metrics.
>
> * We have addressed the typos and omitted Table 2.

---

> > ### Comment · Reviewer_WTWp · 2021-11-29
> > **Comments to authors response**
> >
> > I still consider that the comparisons to ERM and Intra-class mixup are not necessarily fair. Unfortunately, the authors did not comment much on this point in the rebuttal.
> >
> > Despite this, most of the questions that I had raised have been addressed. Thanks to the authors for the detailed response. I will upgrade my rating accordingly.

---

### Official Review · Reviewer_xKKx · 2021-11-02

**Correctness:** 4
**Technical Novelty And Significance:** 3
**Empirical Novelty And Significance:** 3
**Recommendation:** 8
**Confidence:** 4

**Main Review:**

Strengths:

The paper is well written and easy to follow. The method of intra-class mixup makes sense to me. The results (including both tables and figures) are clearly presented.


Weakness:

The results show that adding Cos(\theta) on top of the model with intra-class mixup is helpful. I wonder if a regular model without intra-class mixup can also get improvement by adding Cos(\theta) to the OOD metric.

The method has been compared with simple baselines MSP, ODIN, and energy score. It would be great if the method can provide supplementary improvement for other recently developed methods such as Bayesian NNs or Gaussian Process.

The method is evaluated mostly on far-OOD benchmarks (see Table 4 and 6, 7, 8). I wonder if the proposed method can work well for near-OOD benchmarks such as CIFAR-10 vs CIFAR-100.


**Summary Of The Paper:**

The paper proposes to use intra-class mixup to train OOD detectors. Adding intra-class mixup, the separability between in-distribution and out-of-distribution data is improved. The methods are evaluated in multiple OOD benchmark datasets, showing improvement of 4-6% over the methods without intra-class mixup.

The paper also interestingly shows that the cosine of the angular margin is also a useful measure for OOD detection. It can be added to other regular OOD measures to further improve the performance.



**Summary Of The Review:**

The paper proposes an interesting idea of using intra-class mixup to improve separability between in- and out-of-distribution data. The paper also shows that cos(\theta) can be a useful measurement for OOD detection, and can be added to other OOD measurements like MSP and ODIN. It would be great if more results can be provided

---

> ### Author Response · Authors · 2021-11-23
> **Authors' response**
>
> We thank the reviewer for their valuable feedback.
>
> *Comment*: The results show that adding Cos(\theta) on top of the model with intra-class mixup is helpful. I wonder if a regular model without intra-class mixup can also get improvement by adding Cos(\theta) to the OOD metric.
>
> *Authors*: Adding Cos(theta) on top of ERM and inter-class mixup trained models does improve their performance. We have added the results for ERM and inter-class mixup with Cos(theta) in the appendix (Table 12). Intra-class mixup with Cos(theta) performs the best when compared with ERM or inter-class mixup with Cos(theta).
>
>
> *Comment*: The method has been compared with simple baselines MSP, ODIN, and energy score. It would be great if the method can provide supplementary improvement for other recently developed methods such as Bayesian NNs or Gaussian Process.
> The method is evaluated mostly on far-OOD benchmarks (see Table 4 and 6, 7, 8). I wonder if the proposed method can work well for near-OOD benchmarks such as CIFAR-10 vs CIFAR-100.
>
> *Authors*: We have added the result for near-OoD (CIFAR-10 vs CIFAR-100) in the appendix (Table 11). And this seems to favor inter-class mixup.

---

### Official Review · Reviewer_Fzqo · 2021-11-02

**Correctness:** 3
**Technical Novelty And Significance:** 3
**Empirical Novelty And Significance:** 3
**Recommendation:** 6
**Confidence:** 3

**Main Review:**

Strengths:

Using intra-class mixup to increase angular separability between in-distribution and OoD and hence OoD performance.

Adding angular margin to the OoD scores improves OoD performance.

Weaknesses:

The improvement in performance is not significant.

Table 3: the purpose and analysis of Table 3 could be further discussed.   The "baseline accuracy" on In-distribution and/or OoD data?  It seems to be in-distribution data.  Accuracy of Intra-mixup is lower in 3 out of 4 datasets, any suggesions on reasons?

Equation 8: theta_o and theta_i -- are they averages of all the samples in the in-distribution and OoD datasets?


**Summary Of The Paper:**

The authors propose adding angular distance in the OoD score and additional training samples via intra-class mixup.  Angular margin/distance is defined as the angle between the weight vector from a class and an instance (ie. arccos of the dot product of the unit weight vector and the unit vector for a instance.  Intra-class mixup generates samples by interpolating between two samples of the same class.   Angular spread is the standard deviation of the angular margin for a given dataset.  Intra-class mixup reduces angular spread in in-distribution data and increases angular spread between in-distribution and OoD data.   Angular separability is defined as the squared difference of in-distribution angular margin and OoD angular margin, normalized by the sum of their standard deviation.

With four datasets, they show that intra-class mixup increases angular separability in 3 datasets.  With four datasets and 3 different OoD techniques, intra-mixup is more accurate in AUROC and FPR95, but not AUPR.   Adding the cosine of angular margin in the OoD scores generally improves performance over using OoD scores alone.





**Summary Of The Review:**

The ideas of using intra-class mixup to increase angular separability and adding angular margin to the OoD scores are interesting.

However, the presentation can be improved.

---

> ### Author Response · Authors · 2021-11-23
> **Authors' response**
>
> We thank the reviewer for their valuable feedback.
>
> *Comment*: Equation 8: theta_o and theta_i -- are they averages of all the samples in the in-distribution and OoD datasets?
>
> *Authors*: theta_o is the average over OoD datasets and theta_i is the average over the in-distribution dataset.
>
> *Comment*: Table 3: the purpose and analysis of Table 3 could be further discussed. The "baseline accuracy" on In-distribution and/or OoD data? It seems to be in-distribution data. Accuracy of Intra-mixup is lower in 3 out of 4 datasets, any suggesions on reasons?
>
> *Authors*: We thank the reviewer for the feedback. The baseline accuracy is on the in-distribution dataset. Baseline accuracy tends to have some effect on the results. For example, on TinyImageNet lower baseline accuracy is partially responsible for lower AUROC and AUPR scores. Thus, this is an important reference point.
>
> Please note that intra-class mixup has better accuracy than ERM, while it is slightly worse than inter-class mixup. The improved performance on inter-class mixup has been attributed to the soft labels and the regularization effect it has. Intra-class mixup can be viewed as a hybrid between inter-class mixup and ERM. While intra-class mixup has benefits for OoD detection, intra-class mixup reduces the effect of soft labels thus reducing the regularization effect leading to the lower performance on baseline compared to inter-class mixup.

---

> > ### Comment · Reviewer_Fzqo · 2021-11-25
> > **comments on author response**
> >
> > Thanks for providing clarification.
> >
> > Table 2 on baseline accuracy seems to be not discussed in the text.  Further discussion between Table 2 and Table 3 would be nice if the baseline accuracy has an impact on AUROC, ... performance.

---

### Decision · Program_Chairs · 2022-01-20

**Decision:**

Reject

**Comment:**

The paper proposes to use intra-class mixup supplemented with angular margin to improve OOD detection.

Strengths:
+ Simple idea
+ Experiments on multiple datasets (although mostly focused on image benchmarks)

Weaknesses:
- Justification for the idea could be improved. It'd be nice to understand when we expect this to (not) work.
- Differences from prior work "Angle-based outlier detection in high-dimensional data" could be better explained.

While the paper has some interesting contributions, the reviewers and I feel that the current version falls short of the acceptance threshold. I encourage the authors to revise and resubmit to a different venue.